# Ozone-Gravity Wave Interaction in the Upper Stratosphere/Lower Mesosphere

Author: Axel Gabriel

*Leibniz-Institute of Atmospheric Physics at the University Rostock e.V. (IAP), Kühlungsborn, Germany*

*Corresponding author*: Axel Gabriel, gabriel@iap-kborn.de

Submitted to Atmospheric Physics and Chemistry (ACP)

# Ozone-Gravity Wave Interaction in the Upper Stratosphere/Lower Mesosphere

Axel Gabriel

Leibniz-Institute of Atmospheric Physics at the University Rostock e.V. (IAP), Kühlungsborn, Germany

*Correspondence to*: Axel Gabriel (gabriel@iap-kborn.de)

**Abstract.** The increase in amplitudes of upward propagating gravity waves (GWs) with height due to decreasing density is usually described by exponential growth. Recent measurements show some evidence that the upper stratospheric/lower mesospheric gravity wave potential energy density (GWPED) increases stronger during daylight than nighttime. This paper suggests that ozone-gravity wave interaction can principally produce such a phenomenon. The coupling between ozone-photochemistry and temperature is particularly strong in the upper stratosphere where the time-mean ozone mixing ratio is decreasing with height; therefore, an initial ascent (or descent) of an air parcel must lead to an increase (or decrease) in ozone and in the heating rate compared to the environment, and, hence, to an amplification of the initial wave perturbation. Standard solutions of upward propagating GWs with linear ozone-temperature coupling are formulated suggesting amplitude amplifications during daylight at a specific level of 5 to 15% for low-frequency GWs (periods ≥4 hours), as a function of the intrinsic frequency which decreases if ozone-temperature coupling is included. Subsequently, the cumulative amplification during the upward level-by-level propagation leads to much stronger GW amplitudes at upper mesospheric altitudes, i.e., for single low-frequency GWs, up to a factor from 1.5 to 3 in the temperature perturbations and 3 to 9 in the GWPED increasing from summer low to polar latitudes. Consequently, the mean GWPED of a representative range of mesoscale GWs (horizontal wavelengths between 200 and 1100 km, vertical wavelengths between 3 and 9 km) is stronger by a factor from 1.7 to 3.4 (2 Jkg$^{-1}$ to 50 Jkg$^{-1}$, or 2% to 50% in relation to the observed order of 100 Jkg$^{-1}$, assuming initial GW perturbations of 1 K to 2 K in the middle stratosphere). Conclusively, the identified process might be an important component in the middle atmospheric circulation, which is not considered up to now.

## 1 Introduction

Atmospheric gravity waves (GWs), with horizontal wavelengths of 100 km to 2000 km, are produced in the troposphere and propagate vertically through the stratosphere and mesosphere, where gravity wave breaking processes are an important driver of the middle atmospheric circulation (e.g., Andrews et al., 1987; Fritts and Alexander, 2003). Usually, upward propagating GWs are described by sinusoidal wave perturbations in a slowly varying background flow with an exponentially growing amplitude with height due to decreasing density ($\sim e^{z/2H}$, where H is the scale height). Recently, Baumgarten et al. (2017) found some evidence that the growth of the GW amplitudes between middle stratosphere and mesosphere might be

stronger during daylight than nighttime. The aim of the present paper is to examine whether ozone-gravity wave interaction can principally produce such an amplification.

Seasonal variations of gravity wave potential energy density (GWPED) have been derived based on satellite data or lidar measurements (e.g., Geller et al., 2013; Ern et al., 2004; Kaifler et al., 2015, Baumgarten et al., 2017; Ern et al., 2018). At summer mid- and polar latitudes, the order of the monthly mean GWPED increases from approximately 1 $Jkg^{-1}$ in the middle stratosphere (30-40 km) to 10 $Jkg^{-1}$ in the lower mesosphere (50-60 km) and 100 $Jkg^{-1}$ in the upper mesosphere (80-90 km), with usual initial GW perturbations in the middle stratosphere in the order of about 1 K to 2 K, and wave periods primarily between 4 to 10 hours (e.g., Kaifler et al., 2015; Baumgarten et al., 2017, 2018; Ern et al., 2018). Generally, the GW sources in the middle stratosphere are weaker but the relative increase in the GWPED between middle stratosphere and upper mesosphere stronger during summer than winter, including a less pronounced seasonal cycle in the upper mesosphere than in the levels below, which is primarily due to the seasonal change in critical level filtering of the GWs by the zonal wind (e.g., Kaifler et al., 2015; Ern et al., 2018), but also due to specific GWs generated by convection and propagating towards polar latitudes (Chen et al., 2019), or to additional sources of GWs in the mesosphere independent from the GWs at lower levels (Reichert et al., 2021). Recently, model simulations with resolved GWs suggested multistep vertical coupling processes producing such secondary GWs as a result of dissipating primary GWs, which can strongly enhance the GW amplitudes in the upper mesosphere (e.g., Becker and Vadas, 2018; Vadas et al., 2018 a, b). However, the potential role of daylight-nighttime differences in the increase of GW amplitudes with height have been considered only very sparsely up to now.

Baumgarten et al. (2017) derived monthly means of the GWPED from full-day Lidar temperature measurements at northern mid-latitudes (54°N, 12°E), and found a stronger relative increase between 35-40 km and 55-60 km for full-day than nighttime observations during summer months, but less pronounced differences during winter. For example, for July, the GWPED at 55-60 km show values of about $1 \cdot 10^{-2}$ $Jm^{-3}$ (or 10 $Jkg^{-1}$) for full-day measurements but about $0.5 \cdot 10^{-2}$ $Jm^{-3}$ for nighttime only (or 0.2 but 0.1 $Jm^{-3}$, if the measured temperature fluctuations are vertically filtered for vertical wavelengths $L_m < 15$ km), where the GWPED at 35-40 km remains nearly unchanged, indicating a difference between full-day- and nighttime values by a factor of about 2. Generally, measurements of the mesospheric GWPED are much more uncertain during summer than winter months (e.g., Kaifler et al., 2015; Ehard et al., 2015; Baumgarten et al., 2017), and the signal-to-noise ratio of the lidar measurements is less good during daylight than nighttime (e.g., Rüfenacht et al., 2018), which can stimulate some doubt on the reliability of the daytime-nighttime differences derived from these specific measurements. In addition, taking the potential uncertainties of the analyzing methods into account (i.e., the temporal filtering methods used for the measured time series), Baumgarten et al. (2017) speculated that a change in the phase of long periodic waves (e.g., diurnal and semidiurnal tides) could change the filtering conditions for GWs. However, conclusively Baumgarten et al. (2017) assumed that the detected daylight-nighttime differences are of true geophysical origin, where an unequivocal explanation of this phenomenon remained open. Considering also that full-day observations of Baumgarten et al. (2018) during May 2016 showed pronounced GW activity particularly at altitudes between 42 km and 50 km, where the coupling between ozone and temperature is particularly strong, it seems to be worthwhile to examine whether ozone-gravity wave

interaction could principally lead to such daylight-nighttime differences in the GW amplitudes. This must then also lead to a potential effect on the differences in the GWPED between polar day and polar night. The examination of the present paper is based on standard equations describing upward propagating GWs in a constant background flow excluding other processes controlling the GWPED variability to provide clear understanding and quantification of the potential effect, which cannot be achieved based on observational data analysis or comprehensive model calculations alone.

The coupling of temperature and ozone is particularly strong in the upper stratosphere due to the short photochemical lifetime of ozone (e.g., Brasseur and Solomon, 1995). Linear relationships for a change in the heating rate due to a change in ozone, and a change in photochemistry due to a change in temperature, were derived from basic theory or satellite observations, and have been introduced in standard equations of stratospheric dynamics to examine the effects on the stratospheric circulation, planetary-scale wave patterns and equatorial Kelvin waves (Dickinson, 1973; Douglass et al., 1985;

Froidevaux et al., 1989; Cordero et al. 1998, 2000; Nathan et al., 2007; Ward et al., 2010; Gabriel et al., 2011a). Large-scale ozone-dynamic coupling processes show also significant effects in numerical weather prediction or general circulation models (Cariolle and Morcrette, 2006; Gabriel et al., 2007, 2011b; Gillet et al., 2009; Waugh et al., 2009; McCormack et al., 2011; Albers et al., 2013). However, possible effects of mesoscale ozone-gravity wave interaction in the upper stratosphere/lower mesosphere (USLM) have not been considered up to now.

The basic idea of the present paper can be summarized as follows. In the USLM, the time-mean ozone mixing ratio $\mu_0(z)$ is decreasing with height ($\partial\mu_0/\partial z < 0$). Therefore, at a specific level in the ULSM, an ascending air parcel initially forced by an upward propagating sinusoidal GW pattern (i.e., the wave crest with vertical velocity perturbation $w' > 0$) must lead to an increase $\partial\mu'/\partial t > 0$ by both transport (because $-w'\partial\mu_0/\partial z > 0$) and photochemistry (because the temperature-dependent ozone production increases in case of adiabatic cooling), and, hence, in the heating rate $Q'(\mu') > 0$, comparable to the latent heat

release in the troposphere in case of condensation. Then, the induced perturbation $\Delta\theta' > 0$ ($\theta$ is potential temperature) reinforces the initial ascent, where the lapse rate $\partial(\theta_0 + \Delta\theta')/\partial z < \partial\theta_0/\partial z$ decreases ($\partial z = $ constant) suggesting an effective *ozone adiabatic lapse rate* in the upper stratosphere comparable to the *moist adiabatic lapse rate* in the troposphere. Analogously, a descending air parcel (the wave trough where $w' < 0$) leads to a decrease $\partial\mu'/\partial t < 0$ and a corresponding change $Q'(\mu') < 0$, reinforcing the initial descent. Overall, this process must lead to a significant amplification of the initial GW amplitude at

this level, and, hence, to a successive amplification of the amplitude during the upward level-by-level propagation through the ULSM.

In Section 2, standard equations for GWs in a zonal mean background flow with and without linearized ozone-temperature coupling are formulated to quantify the amplitude amplification at a specific level (or altitude) and latitude. Then, in section 3, the cumulative amplitude amplification during the propagation through the USLM is derived, based on an idealized

approach of the upward level-by-level propagation of GWs with specific horizontal and vertical wavelengths. Section 4 concludes with summary and discussion.

## 2. Ozone-gravity wave interaction

In the following, ozone-gravity wave interaction is analysed based on standard equations describing GWs in a background atmosphere, where the solutions are illustrated for southern summer conditions. The background is prescribed by monthly and zonal mean temperature $T_0$, ozone $\mu_0$ and short-wave heating rate $Q_0$ of January 2001 (Figure 1, a-c) derived from a simulation with the high-altitude general circulation and chemistry model HAMMONIA (details of the model are given by Schmidt et al., 2010). The heating rate $Q_0$ (Figure 1c) is primarily due to the absorption of solar radiation by ozone and largely agrees with southern summer solar heating rates derived from satellite measurements by Gille and Lyjak (1986) but with somewhat smaller maximum values (in the order of ~10%). Figure 1c shows that $Q_0$ is particularly strong in the upper stratosphere and lower mesosphere (USLM) where $\partial\mu_0/\partial z < 0$ (the dashed line in Figure 1b indicates $\partial\mu_0/\partial z = 0$). The HAMMONIA model includes 119 layers up to 250 km with increasing vertical resolution between ~0.7 km in the middle stratosphere and ~1.4 km in the middle mesosphere, with a horizontal resolution of 3.75°; in the following, this grid is used to illustrate the analytic solutions of upward propagating GWs.

### 2.1 Amplification of gravity wave amplitudes at a specific level

#### 2.1.1 Basic equations

Following Fritts and Alexander (2001), we consider standard equations (1)-(5) describing gravity wave propagation in a background flow, with linear gravity wave perturbations $T'$, $\theta'$, $u'$, $v'$, $w'$, $p'$ and $\rho'$ ($T'$ is temperature, $\theta'=T'(p_{00}/p)^\kappa$ is potential temperature, $p(z)$ is pressure, $p_{00}=1000$ hPa, z is altitude, $u'$, $v'$ and $w'$ are zonal, meridional and vertical wind perturbations, $p'$ and $\rho'$ are the perturbations in pressure and density). Additionally, we include an ozone-dependent heating rate perturbation $Q'(\mu')$ in the potential temperature equation (Eq. 5), and Eq. (6) for the ozone perturbation $\mu'$ with a temperature-dependent perturbation in ozone photochemistry $S'(T')$, where $a(\varphi,z)>0$ and $b(\varphi,z)>0$ are linear coupling parameters as a function of latitude $\varphi$ and altitude z specified below ($\rho_0(z)=\rho_{00}\,\exp^{-(z-z0)/H}$ is background density, H~7km is scale height, $\rho_{00}$ is a reference value at altitude $z_0$, $u_0$ is a zonal mean background wind, $d_0/dt=\partial/\partial t+u_0\partial/\partial x+v_0\partial/\partial y$ where $\partial/\partial x$ and $\partial/\partial y$ denote the derivations in longitude and latitude, g is the gravity acceleration, f is the Coriolis parameter; the background shear terms $w'\partial u_0/\partial z$ and $w'\partial v_0/\partial z$ are neglected because of the Wentzel-Kramers-Brillouin or WKB approximation):

$$\frac{d_0 u'}{dt} + \frac{1}{\rho_0}\frac{\partial p'}{\partial x} = fv' \tag{1}$$

$$\frac{d_0 v'}{dt} + \frac{1}{\rho_0}\frac{\partial p'}{\partial y} = -fu' \tag{2}$$

$$\frac{d_0 w'}{dt} + \frac{1}{\rho_0} \frac{\partial p'}{\partial z} = g \frac{\theta'}{\theta_0} \tag{3}$$


$$\frac{d_0 \rho'}{dt} + \frac{\partial u'}{\partial x} + \frac{\partial v'}{\partial y} + \frac{1}{\rho_0} \frac{\partial \rho_0 w'}{\partial z} = 0 \tag{4}$$

$$\frac{d_0 \theta'}{dt} + w' \frac{\partial \theta_0}{\partial z} = Q' \left(\frac{p_{00}}{p}\right)^\kappa = \frac{a}{\mu_0} \frac{d_0 \mu'}{dt} \tag{5}$$

$$\frac{d_0 \mu'}{dt} + w' \frac{\partial \mu_0}{\partial z} = S' \qquad = -b \, \mu_0 \frac{d_0 \theta'}{dt} \tag{6}$$

Setting $Q'=0$, the dispersion relation for gravity waves results from Eqs. (1)-(5) by introducing sinusoidal perturbations $X_1'=X_{a0} \cdot \exp[i(k_1 x + l_1 y + m_1 z - \omega_1 t)] \cdot \exp^{(z-zs)/2H}$, where $X_1'$ denotes the perturbation quantities, $X_{a0}$ the initial amplitude at altitude zs at the lower boundary of the upper stratosphere, $\exp^{(z-zs)/2H}$ the exponential growth of the amplitude due to

decreasing density, $k_1$ and $l_1$ the horizontal and meridional wave number, $m_1<0$ the vertical wave number for upward propagating GWs with $|m_1|=2\pi/L_{m1}$ and vertical wavelength $L_{m1}$, and $\omega_1$ the frequency (here, the subscript 1 denotes the solutions for $Q'=0$). We focus on horizontal and vertical wavelengths $L_{h1} \geq 50$ km and $L_{m1} \leq 15$ km, where $k_{h1}=2\pi/L_{h1}$ is the horizontal wave number given by $k_{h1}=(k_1^2+l_1^2)^{1/2}$, therefore $(1+k_{h1}^2/m_1^2) \approx 1$. Compressibility effects due to the vertical change in background density are excluded assuming $m_1^2 \gg 1/4H^2$, which is valid for vertical wavelengths $L_m \leq 30$ km.

Then, the dispersion relation for the intrinsic frequency $\omega_{i1}=\omega_1-k_1 u_0$ is given for the frequency range $N_0^2 > \omega_{i1}^2 > f^2$, where $N_0^2=(g/\theta_0) \cdot \partial\theta_0/\partial z$ denotes the Brunt-Vaisala frequency:

$$\omega_{i1}^2 = \frac{N_0^2 \, k_{h1}^2 + m_1^2 f^2}{k_{h1}^2 + m_1^2} \approx N_0^2 \frac{k_{h1}^2}{m_1^2} + f^2 \tag{7}$$

*2.1.2 Ozone-temperature coupling*

For specifying the parameter b, we consider the vertical ascent $w'_1>0$ in the wave crest of an initial sinusoidal GW perturbation, related to an adiabatic cooling term $d_0\theta'_1/dt=-w'_1 \cdot \partial\theta_0/\partial z<0$, which leads to an initial ozone perturbation $\mu'_1>0$ due to the induced increase $d_0\mu'_1/dt=-w'_1 \cdot \partial\mu_0/\partial z>0$ via transport, and to a change in ozone photochemistry described by $S'(T'_1)$ (for the descent $w'_1<0$ in the wave trough, the formulations are analogously but with $\mu'_1<0$ and

$d_0\theta'_1/dt = -w'_1 \cdot \partial\theta_0/\partial z > 0$). In the USLM region, ozone is very short lived and approximately in photochemical equilibrium (Brasseur and Solomon, 1995), i.e., for pure oxygen chemistry it is approximately given by

$$O_3 = \left(\frac{k_2}{k_3} M(O_2)^2 \frac{J_2(O_2)}{J_3(O_3)}\right)^{1/2} \tag{8}$$

where $J_2(O_2)$ and $J_3(O_3)$ are photo-dissociation rates, and $k_2 = 6.0 \cdot 10^{-34} \cdot (300/T)^{2.3}$ cm$^6$s$^{-1}$ and $k_3 = 8.0 \cdot 10^{-12} \cdot \exp(-2060/T)$ cm$^3$s$^{-1}$ chemical reaction rates for ozone production, $O + O_2 + M \rightarrow O_3 + M$, and ozone loss, $O + O_3 \rightarrow 2O_2$ (Appendix C of Brasseur and Solomon, 1995; Table 2 of Schmidt et al., 2010). Accordingly, following Brasseur and Solomon (1995), a relative change in ozone $\Delta\mu_T/\mu_0 = \Delta O_3/O_3$ due to a change in temperature $\Delta T$ is given by

$$\frac{\Delta\mu_T}{\mu_0} = \frac{1}{2}\frac{\Delta(k_2/k_3)}{(k_2/k_3)} = -\frac{1}{2}\left(\frac{2.3}{T_0} + \frac{2060}{T_0^2}\right)\Delta T \equiv -b_0(T_0)\,\Delta T \tag{9}$$

Then, defining $b = b_0 \cdot (p/p_{00})^\kappa$ and introducing a total temperature change $\Delta T/\Delta t$ within a background flow described by $d_0T'/dt = (p/p_{00})^\kappa \cdot d_0\theta'/dt$, the change $S'$ is given by

$$S' = \frac{\Delta\mu_T}{\Delta t} = \frac{\Delta\mu_T}{\Delta T}\frac{\Delta T}{\Delta t} = -\mu_0\, b\, \frac{d_0\theta'}{dt} \tag{10}$$

which is the right-hand term of Eq. (6). Overall, the initial ascent $w'_1 > 0$ leads to an increase in ozone via transport, and the related adiabatic cooling to an increase in ozone because of the induced change $S' > 0$; analogously, the initial descent $w'_1 < 0$ leads to a decrease in ozone via transport and an induced change $S' < 0$. The height-dependence of b is specified by
considering that the ozone photochemistry of the USLM region is related to the spatial structure of $Q_0$, which is characterized by a Gaussian-type height-dependence centered at the maximum of $Q_0$ and rapid decrease with latitude in the extra-tropical winter hemisphere (see Figure 1c). Therefore, b is multiplied with the normalized factor $hz = Q_0/Q_{00}$, where $Q_{00}$ is the averaged profile of $Q_0$ over the summer hemisphere ($b \rightarrow b \cdot hz$, where $hz(z) \approx 1$ in the summer upper stratosphere at the altitude where $Q_0$ reach maximum values). A similar approach of Gaussian-type height-dependence in ozone-temperature
coupling was successfully used by Gabriel et al. (2011a) to analyze observed planetary-scale waves in the ozone distribution. Following previous works (e.g., Cordero and Nathan, 1998, 2000; Nathan et al., 2007; Ward et al., 2010; Gabriel et al., 2011a), the sensitivity of the upper stratospheric heating rate to a change in ozone is approximately described by the linear approach $\Delta Q_\mu \approx A \cdot \Delta\mu$, where $A = A(\varphi, z)$ is a time-independent linear function. If we assume the same sensitivity for both the slowly varying background and the mesoscale GW perturbation propagating within the background flow, $Q_0 \approx A \cdot \mu_0$ and
$Q' \approx A \cdot \mu'$, we may write $\Delta Q_\mu/\Delta\mu = Q_0/\mu_0 = Q'/\mu'$. At a specific altitude z or pressure level p(z), we consider a GW perturbation

over the vertical scale of a vertical wavelength, $\Delta z = L_m$. Then, considering that $\partial\mu'/\partial z = im\mu' = (\tau_i/L_m)\cdot(-i\omega_i\mu')$ with $\tau_i = 2\pi/\omega_i$, the first-order heating rate perturbation is given by

$$Q' = L_m \frac{\partial Q'}{\partial z} \approx L_m \frac{\Delta Q_\mu}{\Delta\mu} \frac{\partial\mu'}{\partial z} = \tau_i \frac{Q_0}{\mu_0} \frac{d_0\mu'}{dt} \tag{11}$$


which is the right-hand side of Eq. (5) when defining $a_0 = \tau_i Q_0$ and $a = a_0\cdot(p_{00}/p)^\kappa$. Except in polar summer regions, the effect of $Q'$ is limited by the length of daylight (here denoted by $\tau_{day}$) in case of large wave periods; therefore, we set the time increment to $\tau_i = \tau_{day}$ in case of $\tau_i > \tau_{day}$, which reduces the effect of $Q'$ during the time period of 24 hours (e.g., $\tau_i \leq 12$ hours over the equator). Overall, assuming again an initial ascent $w'_1 > 0$, the induced increase in ozone $\mu' > 0$ at a pressure level $p(z)$

leads to a heating rate perturbation $Q' > 0$ at this level counteracting to the initial adiabatic cooling and therefore reinforcing the initial ascent. Analogously, an initial descent $w'_1 < 0$ is reinforced by inducing a perturbation $Q' < 0$.

Note here that the use of $\Delta z = L_m$ in Eq. (11) provides a suitable measure of the effect of ozone-temperature coupling on the GW amplitudes at a specific level over the vertical distance $L_m$. It is also possible to set a smaller vertical scale $\Delta z < L_m$ leading to smaller values $Q'_{\Delta z} = (\Delta z/L_m)\cdot Q'$ at a specific level, where $\Delta z$ denotes, for example, the distances of a vertical grid

used in a numerical model; this modification does not change the effect over the vertical distance $L_m$ but it provides better vertical resolution when calculating the cumulative amplitude amplification during the upward level-by-level propagation particularly in case of small vertical wavelengths or small vertical group velocities, as described in the next subsection.

### 2.1.3 Amplification of GW amplitudes at a specific level

The parameterizations of $Q'$ and $S'$ provide a useful modification of the potential temperature tendency when introducing $d_0\mu'/dt$ of Eq. (6) into (Eq. 5):

$$(1+ab)\frac{d_0\theta'}{dt} + w'\left(\frac{\partial\theta_0}{\partial z} + \frac{a}{\mu_0}\frac{\partial\mu_0}{\partial z}\right) = 0 \tag{12}$$

Here, the amplification factor $1+ab$ (with $ab > 0$) describes the feedback of the GW-induced ozone perturbation to the change in potential temperature, and $\partial\theta_0/\partial z + (a/\mu_0)\cdot\partial\mu_0/\partial z$ an *ozone adiabatic lapse rate* which is – in the USLM region – smaller than $\partial\theta_0/\partial z$ because of $\partial\mu_0/\partial z < 0$. Alternatively, we may write:

$$\frac{d_0}{dt}\left(\frac{g}{\theta_0}\theta'\right) + N_\mu^2 w' = 0 \tag{13}$$


with

$$N_\mu^2 = \frac{N_0^2 + N_c^2}{(1+ab)} \tag{14}$$

where $N_c^2 = (g/\theta_0) \cdot (a/\mu_0) \cdot \partial \mu_0 / \partial z$. Like for the lapse rate, $N_\mu^2$ is smaller than $N_0^2$ because $N_c^2 < 0$ and $(1+ab) > 1$. If ozone-temperature coupling becomes weak, below and above the USLM region, $N_\mu^2$ converges to $N_0^2$.

Analogously to the standard solution given above, we introduce sinusoidal GW perturbations of the form $X_2' = X_{\mu 0} \cdot \exp[i(k_2 x + l_2 y + m_2 z - \omega_2 t)] \cdot \exp^{(z-zs)/2H}$ in Eqs. (1)-(4) and (13) (here, the subscript 2 denotes the solutions with ozone-gravity wave coupling) which leads to the modified dispersion relation

$$\omega_{i2}^2 = \frac{N_\mu^2 k_{h2}^2 + m_2^2 f^2}{k_{h2}^2 + m_2^2} \approx N_\mu^2 \frac{k_{h2}^2}{m_2^2} + f^2 \tag{15}$$

where $\omega_{i2} = \omega_2 - k_2 u_0$ and $k_{h2} = (k_2^2 + l_2^2)^{1/2}$.

Eq. (13) provides an evident measure of the amplification of a GW amplitude at a specific altitude z or pressure level p(z). On the one hand, introducing the same initial adiabatic potential temperature perturbation $d\theta_1'/dt$ either with or without ozone-temperature coupling leads to $w_2' = w_1' \cdot (N_0^2 / N_\mu^2)$. Consistently, introducing the same initial perturbation $w_1' N_0^2$ leads to $d\theta_2'/dt = d\theta_1'/dt$ or $-i\omega_{i2}\theta_2' = -i\omega_{i1}\theta_1'$. Then, combining $-i\omega_{i2}\theta_2' = -N_\mu^2 w_2'$ and $-i\omega_{i1}\theta_1' = -N_0^2 w_1'$ suggests that the amplitude $\theta_\mu = \theta_{\mu 0} \cdot \exp^{(z-zs)/2H}$ is stronger than $\theta_a = \theta_{a0} \cdot \exp^{(z-zs)/2H}$ by the factor $\omega_{i1}/\omega_{i2} = N_0^2/N_\mu^2 \geq 1$:

$$\theta_\mu = \theta_a \cdot (\omega_{i1}/\omega_{i2}) \tag{16}$$

Overall, the introduced process of ozone-temperature coupling leads to a decrease in the GW frequency and a corresponding amplification in the GW amplitude described by the factor $\omega_{i1}/\omega_{i2}$ or $N_0^2/N_\mu^2$. Note that vertical variations in $N_0^2$ could affect the increase in amplitude with height particularly in the summer upper mesosphere; therefore, $N_0^2$ is vertically averaged over the USLM region (from 30 hPa to 0.03 hPa, or ~25 km to ~70 km altitude) to focus on the effects of ozone-gravity wave interaction only. Note also that the relation $\omega_{i1}/\omega_{i2} = N_0^2/N_\mu^2$ implies not only a change in amplitude but also a slight change in the relation of horizontal and vertical wavenumber described by $(k_{h2}/m_2) = (N_\mu^2/N_0^2)(k_{h1}/m_1) + f^2(N_\mu^2 - N_0^2)/(N_0^2 \cdot N_0^2)$, i.e., a slight change in the direction of upward propagating GWs which is perpendicular to the angle $\alpha$ of the phase lines defined by $\cos(\alpha) = \pm(k_h/m)$. However, as illustrated in the following, ozone-gravity wave interaction is particularly relevant for a range of wavelengths and periods where the induced changes in $\alpha$ are very small (for $L_{m1}/L_{kh1} < 0.05$, or wave periods $\tau_i > 2h$, the change in $\alpha$ is less than $1 \cdot 10^{-4}$ degree).

*2.1.4 Examples for the amplification of GW amplitudes at specific levels*

Figure 1d-f shows the factor $1+ab$ and the quotient $N_0^2/N_\mu^2$ for a GW with horizontal and vertical wavelengths $L_k=500$ km and $L_m=5$ km, and the quotient $N_0^2/N_\mu^2$ for a GW with $L_k=800$ km and $L_m=3$ km. In the first example, the factor $1+ab$ (Figure 1d) contributes to the amplification of the GW amplitude at a specific level by up to 6-8%, and the overall factor $N_0^2/N_\mu^2=(1+ab)\cdot N_0^2/(N_0^2+N_c^2)$ (Figure 1e) by up to 8-12% (including a decrease in the lapse rate of up to 3% described by $(N_0^2+N_c^2)/N_0^2$, here not shown). The second example (Figure 1f) shows that the factor $N_0^2/N_\mu^2$ is larger in case of larger horizontal and smaller vertical wavelength, reaching amplifications of up to 12% to 14% (shaded areas denote the latitudinal range where the amplification is reduced due to the length of daylight, i.e., where $\tau_i>\tau_{day}$).

For illustration of the induced change in ozone at a specific level (Figure 2 a-d), we assume an initial GW perturbation $\theta'_1$ with exponentially growing amplitude $\theta_a=\theta_{a0}\cdot\exp^{(z-zs)/2H}$, with an initial temperature amplitude $T_{a0}$ of 1 K at $zs\approx35$ km ($ps=6.28$ hPa) increasing to $\sim8$ K at $z\approx65$ km ($p=0.1$ hPa). In the present paper, we formulate the solutions for pressure levels p, i.e., the initial perturbation is alternatively described by $\theta_a=\theta_{a0}\cdot(ps/p)^{1/2}$ assuming $p=ps\cdot\exp^{(z-zs)/H}$. Introducing the associated perturbation $w'_1=-(\partial\theta_0/\partial z)^{-1}\cdot d_0\theta'_1/dt$ in Eq. (6) leads to $d_0\mu'_1/dt=[(\partial\mu_0/\partial z)/(\partial\theta_0/\partial z)-b\mu_0]\cdot d_0\theta'_1/dt$, and, considering $d_0\mu'_1/dt=-i\omega_{i1}\mu'_1$ and $d_0\theta'_1/dt=-i\omega_{i1}\theta'_1$, to an initial ozone perturbation $\mu'_1=\theta'_1\cdot[(\partial\mu_0/\partial z)/(\partial\theta_0/\partial z)-b\mu_0]$. For the example of the ascent ($w'_1>0$) shown in Figure 2, we set $\theta'_1<0$ leading to $\mu'_1>0$. For $L_k=500$km and $L_m=5$km, the contributions $\mu'(TR)=\theta'_1\cdot[(\partial\mu_0/\partial z)/(\partial\theta_0/\partial z)]$ related to transport (Figure 2a) and $\mu'(CH)=-b\mu_0\theta'_1$ related to $S'$ (Figure 2b) sum up to a total change of $\mu'\approx0.2$ to 0.5 ppm (Figure 2c) or $\mu'/\mu_0\approx5$ to 10 % (Figure 2d) in the USLM region where the feedback to the heating rate is particularly strong.

The related change in the heating rate at a specific level (Figure 2e) is given by comparing Eq. (5) with and without ozone-temperature coupling. Assuming the same initial ascent or adiabatic cooling as above leads to $(w'_2-w'_1)(\partial\theta_0/\partial z)=Q'(\mu'_1)$, or, when introducing $w'_2=(\omega_{i1}/\omega_{i2})\cdot w'_1$, to $Q'(\mu'_1)=(\omega_{i1}/\omega_{i2}-1)(-\omega_{i1}\theta'_1)=a\omega_{i1}\mu'_1\mu_0^{-1}$ (where $Q'(\mu'_1)>0$ in case of $w'_1>0$). Figure 2e shows that $Q'(\mu'_1)$ reach values of 0.15 K hr$^{-1}$ over the tropics and 0.25 K hr$^{-1}$ at southern summer polar latitudes. Then, consistently with Eq. (16), we yield $\theta'_2-\theta'_1=(\omega_{i1}/\omega_{i2}-1)\cdot\theta'_1$ with $(\omega_{i1}/\omega_{i2}-1)=-a\mu_0^{-1}[(\partial\mu_0/\partial z)/(\partial\theta_0/\partial z)-b\mu_0]$ for the change in the potential temperature perturbation, i.e., changes in temperature of 0.2 to 0.3 K in the USLM region (Figure 2f). In summary, analogously considering the corresponding change for the descent, we yield an increase in the amplitude of the oscillating GW pattern at a specific level by up to 5 to 10 % in ozone and 0.2 to 0.3 K in temperature.

For other initial wavelengths (or associated frequencies), the latitude-height dependence is very similar to those shown in Figure 1 (d-f) and Figure 2, whereas the magnitude of the amplification factor $\omega_{i1}/\omega_{i2}$ becomes smaller in case of increasing vertical and decreasing horizontal wavelengths, or decreasing frequencies, as illustrated in Figure 3 for an altitude where $\omega_{i1}/\omega_{i2}$ reach maximum values (1.156 hPa or $\approx47$ km altitude). Figure 3a shows values of $\omega_{i1}/\omega_{i2}>1.02$ for wave periods of

$\tau_i$>2h steadily increasing with increasing initial period up to values between 1.14 and 1.15. This value is limited, on the one side, because of the increasing duration of nighttime with latitude towards equatorial and northern winter regions (denoted by shaded areas), and, on the other side, because of the increasing Coriolis force in southern summer mid- and polar regions (i.e., because of $\omega_{i1}^2>f^2$).

Consistently, the amplification factor is increasing with decreasing vertical and increasing horizontal wavelength (Figures 3b and 3c show examples for 70°S and 30°S), where the values are limited by the length of daylight in case of small relations $L_m/L_k$ denoting the conditions where $\tau_i>\tau_{day}$ (Figure 3c, shaded area). Figure 3 also indicates that the examples with $L_k$=500 km and $L_m$=5 km (Figure 1e; Figure 2) and $L_k$=800 km and $L_m$=3 km (Figure 1f) represent scales where ozone-gravity wave interaction is particularly efficient.

Overall, Figures 1 (d-f), 2 and 3 illustrate the amplification of GW amplitudes at a specific level and a specific time; as far as the GWs are continuously propagating upward through several levels where $\omega_{i1}/\omega_{i2}-1>0$, the amplification will be successively reinforced at each level. This cumulative amplification can lead to much stronger GW amplitudes at upper mesospheric altitudes in case with than without ozone-gravity wave interaction as demonstrated in the next subsection.

## 2.2 Upward propagating GWs in a background flow

### 2.2.1 Level-by-level amplification of GW amplitudes

In the following, a solution of the cumulative amplification during the vertical level-by-level propagation is derived, excluding – to a first guess – other effects like small-scale diffusion, wave breaking processes, interaction of the GWs with atmospheric tides, or so-called secondary gravity waves. Following Huygens principle, each point of a propagating wave front at a specific level is the source of a new wave at this level, i.e., a single upward propagating GW, which is amplified at a level $z_{j-1}$, is the initial perturbation amplified at the next level $z_j$. For illustration (Figure 4, a-c), we choose an initial GW with horizontal and vertical wavelengths $L_m$=500 km and $L_m$=5 km as above, where the vertical distance between the levels $z_{j-1}$ and $z_j$ is set by the initial vertical wavelength $\Delta z=L_m$. First, we focus on polar latitudes during southern polar summer (70°S) with daylight conditions only, then we consider the modification for mid- and equatorial latitudes where GWs with weak vertical group velocities propagate through the USLM during both daylight and nighttime.

For orientation, Figure 4a shows the profiles $\omega_{i1}/\omega_{i2}$ for $L_k$=500 km and $L_m$=5 km at 70°S (solid), and, for comparison, for $L_m$=3 km (dashed) and $L_m$=9 km (dotted), indicating the altitude range where ozone-temperature coupling is relevant (note that the depicted distance of pressure levels represents approximately a 5 km distance in altitude). Beginning with a first level at zs≈35 km (6.28 hPa), the wave propagates through 8 layers between ≈35 km and ≈70 km (0.06 hPa) where the amplification of the amplitude is relevant. At each of these levels, denoted by $z_j=zs+(j-1)\cdot\Delta z$ (j=1, n; here n=8), the amplitude at $z_j$ will be amplified by the factor $\omega_{i1}(z_j)/\omega_{i2}(z_j)$ at $z_j$. Starting with an exponentially growing amplitude $T_a(z)=T_a(zs)\cdot\exp^{(z-zs)/2H}$ (where we set again $T_a(zs)$=1 K), we yield a new amplitude $T_{a1}(z_1)=T_a(z_1)\cdot\omega_{i1}(z_1)/\omega_{i2}(z_1)$ at the level

$z_1$ defining a new exponentially growing amplitude $T_{\mu 1}(z)=T_{a1}(z_1)\cdot\exp^{(z-z1)/2H}$. Then, we yield $T_{a2}(z_2)=T_{\mu 1}(z_2)\cdot\omega_{i1}(z_2)/\omega_{i2}(z_2)$ at the level $z_2$ defining $T_{\mu 2}(z)=T_{a2}(z_2)\cdot\exp^{(z-z2)/2H}$, and so on. Finally, the amplitude at the level $z_n$ in the middle mesosphere is described by

$$T_{\mu n}(z)=T_a(z)\cdot\prod_{j=1}^{n}\left[\frac{\omega_{i1}(z_j)}{\omega_{i2}(z_j)}\right]\ ,\tag{17}$$


where the product symbol $\Pi_{j=1,\,n}$ denotes the multiplication with $\omega_{i1}(z_j)/\omega_{i2}(z_j)$ at each level $z_1\le z_j\le z_n$. As mentioned above, the solutions are calculated on pressure levels, i.e., $z$ represents the geopotential height, and the vertical distance $\Delta z$ between the levels is given by $\Delta z=-(\rho_0 g)^{-1}\Delta p=-H(T_0)\cdot(\Delta p/p)$, where $H(T_0)=g/(RT_0)$ is the height-dependent scale height defined by the background; note here that using a constant scale height $H_0=7$ km instead of $H(T_0)$ leads only to second-order changes in

the cumulative amplitude amplification (the sensitivity test is described below in Section 2.2.4), because $H(T_0)$ is varying only slightly in the USLM region (between ~7.5 km at summer stratopause altitudes and ~6.5 km at 70 km).

Figure 4b shows the initial amplitude $T_a$ (blue line) and the series of the successively amplified amplitudes $T_{\mu 1}$, $T_{\mu 2}$, …, $T_{\mu n}$ (from light blue towards red line), and Figure 4c the related series of constant relative values $T_{\mu 1}/T_a$, $T_{\mu 2}/T_a$, …, $T_{\mu n}/T_a$ starting at the level $z_j$ (solid lines) together with the previous values starting at $z_{j-1}$ multiplied by the factor $\omega_{i1}/\omega_{i2}$ (dotted

lines), illustrating the successively increasing growth of the amplitude during the upward level-by-level propagation. Finally, the amplitudes converge to $T_{\mu n}(z)$ when reaching the upper mesosphere, where $T_{\mu n}(z)$ is stronger than $T_a(z)$ by a factor of ~1.47. Figure 4c also shows the fitted relative increase of the amplitude $T_\mu/T_a$ (thick red line) describing the continuous change in the growth rate of the amplitude, where $T_\mu(z)$, or $T_\mu(p)$, is defined by

$$T_\mu(p) = hs(p)\cdot T_a(p) + hm(p)\cdot T_{\mu n}(p)\tag{18}$$

with weighting functions $hs=p_0^{1.5}/(p_0^{1.5}+p_m^{1.5})$ and $hm=1-hs$, where $p_0$ is the background pressure and $p_m(70°S)\approx0.96$ hPa the level of the maximum of $\omega_{i1}/\omega_{i2}$ (note that the height of this maximum is slightly decreasing from $p_m\approx0.89$ hPa over the south pole to $p_m\approx1.3$ hPa over the equator).

For mid- and equatorial latitudes, daylight-nighttime conditions are considered by setting the amplification factor to $F_d=\omega_{i1}/\omega_{i2}$ during daylight but to $F_d=1$ during nighttime over the vertical wave propagation distance of one full day. In detail, we define the parameter $L_{day}=(\tau_{day}-0.5\cdot\tau_0)/(0.5\cdot\tau_0)$, where $\tau_0=24$ hours and $\tau_{day}$ is the duration of daylight within 24 hours at the latitude $\varphi$ (with $L_{day}=1$ during polar summer and $L_{day}=0$ at the equator). Further, considering the vertical group velocity $c_{gz}=\partial\omega_{i1}/\partial m_1=-(\omega_{i1}/m_1)\cdot(\omega_{i1}^2-f^2)/\omega_{i1}^2$ (with initial frequency $\omega_{i1}$ and vertical wavelength $m_1$ as first guess), the sinusoidal

wave propagation structure between the middle stratosphere and middle mesosphere is described by $L_{cgz}=\cos(2\pi\tau_0\cdot(z-z_m)/c_{gz})$

changing periodically between 1 and -1 over one wavelength, where $z$ and $z_m$ are given in km and $c_{gz}$ in km hr$^{-1}$, and where $L_{cgi}=1$ at the level $p_m$, or altitude $z_m(p_m)$. Then, the combined parameter $L_d=L_{day}+L_{cgi}$ separates the vertical propagation distance into daylight and nighttime fractions by defining a constant value $C_d=1$ in case of $L_d>1$ and $C_d=0$ in case of $L_d\leq1$, where the factor $F_d=1+C_d\cdot((\omega_{i1}/\omega_{i2})-1)$ provides $F_d=\omega_{i1}/\omega_{i2}$ in case of daylight and $F_d=1$ in case of nighttime.

As an example, Figure 4d shows the profile of the resulting amplification factor $F_d$ at 10°S for a GW with $L_k=500$ km and $L_m=5$ km as above, with an associated vertical group velocity $c_{gz}$ of about 7 km per 12 hours, illustrating that we define $F_d(z_j)=\omega_{i1}(z_j)/\omega_{i2}(z_j)$ where $z_j$ is located in the daylight region (red) but $F_d(z_j)=1$ where $z_j$ is located in the nighttime region (blue). The indicated vertical wave propagation distance during daylight increases towards southern summer polar latitudes but decreases towards northern winter polar latitudes. Note here that, for vertical wavelengths examined in the present paper

($L_m \leq 15$ km), a vertical shift of the phase – as defined by the altitude $z_m$ in the definition of $L_{cgz}$ – does not have a significant impact on the cumulative amplification of the GW amplitudes because of the Gaussian-type structure of the profile of $F_d=\omega_{i1}/\omega_{i2}$, which has been verified by several test calculations with other levels than $p_m$, or other altitudes than $z_m$.

In the following, the fitted profiles $T_\mu$ are used for further examinations with different horizontal and vertical wavelengths, where the vertical level-by-level amplification is calculated by using the distances $\Delta z=\Delta z_H$ of the vertical grid of

HAMMONIA instead of $\Delta z=L_m$. This includes a smaller amplification factor $F_\omega=\omega_{i1}/\omega_{i2}$ over the vertical distance $\Delta z_H$ because of the smaller heating rate perturbation $Q'_{\Delta zH}=(\Delta z_H/L_m)\cdot Q'$ (see Eq. (11 and related discussion); however, the resulting difference in the amplification at a specific level over the vertical distance $L_m$ are nearly the same except some small differences of less than 0.5% due to the different vertical resolution (i.e., $F_\omega(\Delta z=L_m)\approx1+(F_\omega(\Delta z=\Delta z_H)-1)\cdot(L_m/\Delta z_H)$). Also the resulting cumulative amplification in the upper mesosphere remains nearly unchanged ($T_{\mu nh}(\Delta z=L_m)\approx T_{\mu nh}(\Delta z=\Delta z_H)$,

where nh is the number of the HAMMONIA levels in the USLM), where small differences between $T_{\mu nh}$ and $T_{\mu n}$ of less than 10% occur only at mid- and equatorial latitudes in case of small vertical wavelengths (or small vertical group velocities) when considering the vertical propagation during both daylight and nighttime described below.

*2.2.2 Cumulative amplitude amplification for representative examples*

Figure 4e illustrates the dependence of the amplitude amplification on the horizontal and vertical wavelengths $L_k$ and $L_m$ at 70°S, where it is not affected by nighttime conditions. In comparison to the example of $L_k=500$ km and $L_m=5$ km leading to a cumulative amplification of ~1.47 (red, solid line), a larger vertical wavelength of $L_m=9$ km leads to a smaller value of ~1.15 (red, dotted line), but a smaller vertical wavelength of $L_m=3$ km to a larger value of ~2.27 (red, dashed line), because the induced increase in the ozone perturbation $\mu'$ produces a heating rate perturbation $Q'$ within a shorter (in case of $L_m=9$

km) or larger (in case of $L_m=3$ km) time increment $\tau_i$. For the same reason, the amplification is generally larger if the horizontal wavelength $L_k$ is larger, e.g., in case of $L_k=800$ km, the final amplification in the upper mesospheric amplitudes

amounts to ~1.22 for $L_m$=9 km (purple, dotted line), ~1.63 for $L_m$=5 km (purple, solid line), and ~2.56 for $L_m$=3 km (purple, dashed line).

The related gravity wave potential energy density (GWPED, here denoted by E) is derived following Kaifler et al. (2015):


$$E = \frac{1}{2}\left(\frac{g}{N}\right)^2 \left(\frac{T'}{T_0}\right)^2 \tag{19}$$

Introducing $T'=T_2'$ and $N=N_\mu$, or $T'=T_1'$ and $N=N_0$, leads to the case with ($E_\mu$) or without ($E_a$) ozone-gravity wave interaction. Figure 4f shows the relative amplitudes $E_\mu/E_a$ related to Figure 4e. In case of $L_k$=500 km (red lines), the final

amplification reach values of ~1.32 for $L_m$=9 km (dotted), ~2.17 for $L_m$=5 km (solid), and ~5.21 for $L_m$=3 km (dashed), and in case of $L_k$=800 km (purple) values of ~1.50 for $L_m$=9 km (dotted), ~2.70 for $L_m$=5 km (solid), and ~6.62 for $L_m$=3 km (dashed). Overall, these factors provide a first-order estimate of the effect of ozone-gravity wave coupling at 70°S during polar summer, i.e., in case of large horizontal ($\geq$ 500 km) and small vertical ($\leq$ 5 km) wavelengths, we find cumulative amplifications in the upper mesosphere in the order of ~1.5 to ~2.5 in the temperature perturbations and in the order of ~3 to

~7 in the related GWPED.

*2.2.3 Cumulative amplitude amplification depending on latitude*

For the GW with $L_k$=500 km and $L_m$=5 km, Figure 5 shows the latitudinal dependence of the cumulative amplification of the temperature perturbation (indicated by $T_\mu/T_a$, Figure 5a) and the related GWPED (indicated by $E_\mu/E_a$, Figure 5b). The values

decrease from $T_\mu/T_a\approx$1.5 and $E_\mu/E_a\approx$2.4 over southern summer polar latitudes towards $T_\mu/T_a\approx$1.2 and $E_\mu/E_a\approx$1.4 at lower mid-latitudes (40°S), and then less rapidly towards $T_\mu/T_a\approx$1.1 and $E_\mu/E_a\approx$1.2 at 20°N. Overall, although the amplification of the GW amplitudes decreases rapidly with the decrease in the length of daylight, it is still quite strong at mid-latitudes.

Figure 6 shows the relations $T_\mu/T_a$ (Figure 6a) and $E_\mu/E_a$ (Figure 6b) at upper mesospheric levels (0.01 hPa, ~80 km) for different horizontal and vertical wavelengths as used for Figures 4e and 4f. For both $L_k$=500 km (red) and $L_k$=800 km

(purple), the amplifications of the temperature perturbations and of the related GWPED are strongest for $L_m$=3 km (dashed lines), at polar latitudes with values between 2.5 to 3 in $T_\mu/T_a$ and 7 to 9 in $E_\mu/E_a$, and at mid- and equatorial latitudes between 1.5 to 1.8 in $T_\mu/T_a$ and 2.4 to 3.5 in $E_\mu/E_a$. These values decrease with increasing vertical wavelength, i.e., when changing to $L_m$=5 km (solid lines) or $L_m$=9 km (dotted lines) roughly to ~1.7 or ~1.25 in $T_\mu/T_a$ and ~3.0 or ~1.5 in $E_\mu/E_a$ at polar latitudes, and roughly to ~1.25 or ~1.2 in $T_\mu/T_a$ and ~1.5 or ~1.25 in $E_\mu/E_a$ at mid- and equatorial latitudes. Overall, for

the mesoscale GWs with small vertical and large horizontal wavelengths, the cumulative amplification due to ozone-gravity wave coupling leads to much stronger amplitudes at upper mesospheric altitudes during daylight than nighttime, in the GW

perturbations by a factor between ~1.5 at summer mid-latitudes and ~3 for polar day conditions, and in the GWPED by a factor between ~3 at summer mid-latitudes and ~9 for polar day conditions.

Note here that vertical momentum flux terms $F_{GW}=\rho_0(u'w')$ can be derived from local profiles $T'$ if the background is known,
i.e. by $F_{GW}=\rho_0 E\cdot(k/m)$ (Ern et al., 2004). Therefore, the amplification of the GW amplitudes must lead to the same amplification of the flux term $F_{GW}$ and, if the GWs do not break at lower levels, of the associated gravity wave drag $GWD=-\rho_0^{-1}\partial F_{GW}/\partial z$ in the upper mesosphere, suggesting an important effect of ozone-gravity wave interaction on the meridional mass circulation particularly at polar latitudes. However, more detailed investigations need extensive numerical model simulations with a spectrum of resolved GWs which is beyond the scope of the present paper.

*2.2.4 Sensitivity to varying conditions*

In the following, we estimate the sensitivity of the GW amplitude amplification on non-linear processes and background conditions which could modulate the first-guess results described above. For example, the decrease in the frequency towards $\omega_{i2}<\omega_{i1}$ includes a slight decrease in the vertical group velocity towards $c_{gz2}<c_{gz1}$, which can additionally strengthen the process of amplitude amplification because the wave propagates somewhat more slowly through the ULSM region. However, this effect is at least one order smaller than the first-order process described above as derived from test calculations including this effect. For example, for $L_k=500$ km and $L_m=5$ km, $c_{gz2}$ is smaller than $c_{gz1}$ by 15% to 20% at southern summer polar latitudes and 5% to 10% at mid- and equatorial latitudes. Subsequently, at a specific level, the amplification factor $F_d(c_{gz2})$ is stronger than $F_d(c_{gz1})$ by 2% to 3% at polar latitudes and less than 1% at mid- and equatorial latitudes. Including this change into the successive level-by-level propagation leads to a weak successive increase in the cumulative amplifications by ~5% at 1 hPa to ~10% at 0.01 hPa at polar summer latitudes, and by only ~1% at 1 hPa to ~2% at 0.01 hPa at mid- and equatorial latitudes.

We also estimate the sensitivity of the amplitude amplification on the ozone background $\mu_0$, considering the observed long-term changes in upper stratospheric ozone in the order of up to −8% per decade (e.g., Sofieva et al., 2017; WMO, 2018), and the uncertainty in the maximum of the heating rate $Q_0$ which is smaller in the used HAMMONIA data in the order of ~10% compared to those derived from satellite measurements, as mentioned above. In case of a 10%-reduction in ozone, the cumulative amplification in the upper mesospheric GW amplitudes is weaker by about 5% for the example with $L_m=5$ km and 10% for $L_m=3$ km (i.e., at 70°S, we yield a cumulative amplification of ~1.4 to ~2.25 instead of ~1.5 to ~2.5), and the related amplification of the GWPED is weaker by about 10% for $L_m=5$ km and 20% for $L_m=3$ km (at 70°S, a cumulative amplification of ~2.7 to ~7.2 instead of ~3 to ~9). Analogously, in case of an increase in $Q_0$ by 10%, the cumulative amplification is stronger by 5% or 10% in the GW amplitudes and by 10% or 20% in the related GWPED amplitudes.

Another question arises on the sensitivity of the effect of ozone-gravity wave coupling to atmospheric tides or the diurnal cycle in stratospheric ozone, which are planetary-scale processes changing the background conditions for the propagation of

the mesoscale GW perturbations. For example, Schranz et al. (2018) observed stronger amplitudes in upper stratospheric ozone during daylight than nighttime in the order of 5% (summer solstice) to 8% (May). Such a difference would correspond to a change in the cumulative amplification of the upper mesospheric GW amplitudes or GWPED in the order of 5% to 10% or 10% to 20%, as follows from the sensitivity of the effect on the prescribed long-term change in stratospheric ozone derived above.

Baumgarten and Stober (2019) derived amplitudes of tides in the order of up to 0.5 K in the middle stratosphere (~35 km) increasing up to 2 K at ~50 km and ~4 K at 70 km, which would correspond to a change in the lapse rate in the order of up to 0.1 K km$^{-1}$, or in the Brunt-Vaisala frequency $N_0^2$ in the order of 1%. As follows from Eq. (14), a change in the amplification factor $F_d=N_0^2/N_\mu^2$ due to a relative change $\Delta N_0^2/N_0^2$ is given by the factor $[1+(\Delta N_0^2/N_0^2)]/[1+(\Delta N_0^2/N_0^2)(N_0^2/N_\mu^2)(1+ab)^{-1}]$; therefore, for wavelengths $L_k \geq 500$ km and $L_m \leq 5$ km, a relative increase (decrease) of 1% in $N_0^2$ would lead to a relative decrease (increase) in the amplification factor of up to 0.035% at stratopause altitudes, which is much less than the effects of the changes in the vertical group velocity or in ozone described above. Moreover, even if a relative change $\Delta N_0^2/N_0^2$ would be much larger (10% to 50%), it does not change the amplification factor of a specific level by more than 1% to 3%, and, hence, the cumulative amplification of the GW amplitudes in the upper mesosphere by more than 5 to 10%.

Assuming exponential growth of the amplitudes ($\sim e^{(z-zs)/2H}$) between two levels, the usual approach of a constant scale height (e.g., H~7 km) instead of a height-dependent scale height $H(T_0)=g/(RT_0)$ can principally lead to significant differences in the GWPED profiles (e.g., Reichert et al., 2021). For estimating the relevance of a change in H on the cumulative amplitude amplification, the solutions are also calculated for an initial GW perturbation $\theta_a=\theta_{a0}\cdot \exp^{(z-zs)/2H}$ with a prescribed scale height $H_0=7$ km instead of $\theta_a=\theta_{a0}\cdot(ps/p)^{1/2}$, and a related vertical distance $\Delta z=-H_0\cdot(\Delta p/p)$ instead of $\Delta z=-H(T_0)\cdot(\Delta p/p)$ (note that $H(T_0)$ varies in the USLM region between ~7.5 km at summer stratopause altitudes and ~6.5 km at 70 km). Compared to the values shown in Figures 5 and 6, the cumulative amplification of the upper mesospheric GW amplitudes is weaker by about 5% ($L_m=5$ km) to 10% ($L_m=3$ km) over the summer south pole, and weaker by about 1% ($L_m=5$ km) to 3% ($L_m=3$ km) at summer mid-latitudes; correspondingly, the related GWPED values are weaker by about 7.5% ($L_m=5$ km) to 20% ($L_m=3$ km) over the summer south pole, and 1.5% ($L_m=5$ km) to 5% ($L_m=3$ km) at summer mid-latitudes. Overall, these differences are smaller than the first-order effect of ozone-gravity wave coupling by approximately one order, where the use of $H(T_0)$ instead of $H_0$ at the levels of relevant amplification leads to somewhat stronger amplitude amplifications particularly over the summer south pole, because of the difference between the high background temperatures in the summer stratopause region and the low background temperatures in the summer mesosphere (see Figure 1a).

### 2.2.5 Potential effect on mean GW amplitudes

In the following, the potential effect of ozone-gravity wave interaction is estimated for an average over a representative range of 16 different mesoscale GW events (horizontal wavelengths: 200, 500, 800 and 1100 km, vertical wavelengths: 3, 5,

7 and 9 km; compare with the amplification factor as function of wavelengths shown in Figure b and c). Although these settings are idealistic, the results provide a first-guess quantification of the potential effect on time-mean GWPED values usually derived from measurements, where several different GWs contribute to the analyzed temperature fluctuations derived from the detected temperature profiles.

Figure 7 illustrates both the relative and absolute changes in the resulting mean upper mesospheric GW temperature amplitudes (Figure 7, a, b) and in the mean GWPED (Figure 7, c, d). The relative increase in the mean temperature amplitude (Figure 7 a, solid red line) is stronger by a factor increasing from of about 1.3 ($\pm$0.1) at summer low- and mid-latitudes up to 1.7 ($\pm$0.2) at summer polar latitudes (values in brackets denote the 1-standard deviation). This corresponds to a stronger increase from about 7 K ($\pm$2 K) up to 17.5 K ($\pm$4.5 K) in case of an initial GW perturbation of 1 K in the middle

stratosphere (at 6.28 hPa or $\approx$35 km) (Figure 7 b, solid orange line), and from about 14 K ($\pm$4 K) up to 35 K ($\pm$9 K) in case of an initial GW perturbation of 2 K (Figure 7b, solid purple line).

The relative increase in the mean GWPED (Figure 7 c, solid red line) is stronger by a factor increasing from about 1.7 ($\pm$0.2) at summer low- and mid-latitudes up to 3.4 ($\pm$0.8) at summer polar latitudes. This corresponds to a stronger increase in the absolute GWPED values from about 2 Jkg$^{-1}$ ($\pm$0.5 Jkg$^{-1}$) at summer low- and mid-latitudes up to 12 Jkg$^{-1}$ ($\pm$3 Jkg$^{-1}$) at

summer polar latitudes in case of an initial GW perturbation of 1 K at 35 km (Figure 7 d, solid orange line), and from about 8 Jkg$^{-1}$ ($\pm$2.Jkg$^{-1}$) up to 48 Jkg$^{-1}$ ($\pm$0.5 Jkg$^{-1}$) in case of an initial GW perturbation of 2 K (Figure 7d, solid purple line).

In summary, we find an absolute increase in the order of 7 K to 35 K in the mean GW temperature amplitudes and 2 Jkg$^{-1}$ to 50 Jkg$^{-1}$ in the mean GWPED values assuming usual initial GW perturbations in the order of 1 K to 2 K in the middle stratosphere, where the effect is particularly large during polar day conditions. Note here that, assuming exponential growth

with height only, this potential effect can be much larger in case of stronger initial amplitudes in the middle stratosphere (the absolute changes of the temperature amplitudes increase linearly and those of the GWPED values quadratically with increasing initial GW perturbations at 35 km) and in specific geographical regions or time periods where primary GWs with large horizontal and small vertical wavelengths are excited (e.g., where $L_k \geq 800$ km and $L_m \leq 3$ km). However, the GWs with very large amplitudes might dissipate by non-linear wave breaking processes before reaching the upper mesosphere.

## 3 Summary and conclusions

The present paper shows that ozone-gravity wave interaction in the upper stratosphere/lower mesosphere (USLM) leads to a stronger increase of gravity wave (GW) amplitudes with height during daylight than nighttime, particularly during polar summer. The results include information on both the amplification of the GW amplitudes at a specific level and the

cumulative increase of the amplitudes during the upward level-by-level propagation of the wave from middle stratosphere to upper mesosphere.

In a first step, standard equations describing upward propagating GWs with and without linearized ozone-gravity wave coupling are formulated, where an initial sinusoidal GW perturbation in the vertical ozone transport and temperature-dependent ozone photochemistry produces a heating rate perturbation as a function of the initial intrinsic frequency, which determines the duration of the perturbation at a specific level over the distance of the initial vertical wavelength. The solution reveals an amplification of the ascending and descending perturbations of the sinusoidal GW pattern at this level, i.e., a decrease of the intrinsic frequency due to both the induced changes in the lapse rate (or Brunt-Vaisala frequency) and the positive feedback of the coupling on the initial GW perturbation, and an associated increase of the GW amplitude by a factor $\omega_{i1}/\omega_{i2} \geq 1$ defined by the relation of the intrinsic frequencies without ($\omega_{i1}$) and with ($\omega_{i2}$) ozone-gravity wave coupling. This amplitude amplification is dependent on the horizontal and vertical wavelengths, $L_k$ and $L_m$, where the effect is most efficient for GWs with $L_k \geq 500$ km and $L_m \leq 5$ km, or initial frequencies $\tau_i \geq 4$ hours, representing mesoscale GWs forced by cyclones or fronts, or by the orography of mountain ridges like the Rocky Mountains, Andes or Norwegian Caledonides. For southern summer conditions, strongest amplitude amplifications at specific levels of about 5% to 15% over the perturbation distance of one vertical wavelength are located near the stratopause, with peak values over the equator and over summer polar latitudes.

In a second step, an analytic approach of the upward level-by-level propagation of the GW perturbations with and without ozone-gravity wave interaction reveals the cumulative amplitude amplification, where the wave is propagating upward with the vertical group velocity defined by the initial GW parameters, and where daylight-nighttime conditions at mid- and equatorial regions are considered. Representative examples with different initial wavelengths illustrate that the successive increase of both the GW amplitudes and the related gravity wave potential energy density (GWPED) converge to much stronger amplitudes in the upper mesosphere during daylight than nighttime. This effect is strongly decreasing with latitude between summer polar and mid-latitudes because of the decrease in the length of daylight, nearly constant at equatorial latitudes, and decreasing again with latitude towards insignificant values in the winter extra-tropics.

In summary, the strongest impact of ozone-gravity wave interaction is found for wave periods $\geq 4$ hours (related to the wavelengths $L_k \geq 500$ km and $L_m \leq 5$ km), i.e., in a range of wave periods usually observed at summer mid- and polar latitudes. For prescribed single GWs with large horizontal wavelengths (500 to 800 km) and small vertical wavelengths (3 to 5 km), the upper mesospheric GW temperature amplitudes are stronger by a factor between 1.25 to 1.75 at summer low- and mid-latitudes and 1.5 to 3 at summer polar latitudes, and the corresponding GWPED by a factor between 1.5 to 3.5 and 3 to 9. For a representative range of 16 different mesoscale GW events ($L_k$ between 200 and 1100 km, $L_m$ between 3 and 9 km), the mean temperature amplitudes are stronger by a factor between 1.3 at summer low- and mid-latitudes to 1.7 at summer polar latitudes, e.g., stronger by about 7 K to 17.5 K (or 14 K to 35 K) in case of an initial GW perturbation of 1 K (or 2 K) in the middle stratosphere (at ~35 km). The corresponding relative increase in the mean GWPED is stronger by a factor between 1.7 at summer low- and mid-latitudes and 3.4 at summer polar latitudes, e.g., for the same example as above, stronger by about 2 Jkg$^{-1}$ to 12 Jkg$^{-1}$ (or 8 Jkg$^{-1}$ to 48 Jkg$^{-1}$). These values range in the order between 2% to 50% of the observed order of

the mean upper mesospheric GWPED amplitudes (100 Jkg$^{-1}$). These absolute differences can be larger in case of stronger initial perturbations in the middle stratosphere, or in specific geographical regions or time periods where primary GWs with large horizontal and small vertical wavelengths (e.g., where $L_k \geq 800$ km and $L_m \leq 3$ km) are excited; however, the GWs with very large amplitudes might dissipate by non-linear wave breaking processes before reaching the upper mesosphere. Overall, these values result from an idealistic approach and cannot explain entirely the details of specific measurements; however,

they provide a first-guess quantification of the potential effect of ozone-gravity wave interaction on the GW amplitudes.

The variety of horizontal and vertical wavelengths used in the present paper are representative for mesoscale GWs in the USLM region. Observations suggest characteristic vertical wavelengths of GWs between ~2-5 km in the lower stratosphere increasing to ~10-30 km in the upper mesosphere, but also the existence of large vertical wavelengths greater than 10 km in the ULSM region particularly above convection in equatorial regions or over southernmost Argentina (e.g., Alexander, 1998;

McLandress et al., 2000; Fritts and Alexander, 2003; Hocke et al., 2016; Baumgarten et al., 2018; Reichert et al., 2021). The results of the present paper suggest that the effect of ozone-gravity wave coupling decreases with increasing vertical wavelengths $L_m \geq 9$ km but strongly increases with decreasing vertical wavelengths $L_m \leq 5$ km. The latter could lead to more pronounced gravity wave breaking and dissipation processes in the upper stratosphere during daylight than nighttime, and – subsequently – to more prominent GWs with larger vertical wavelengths of $L_m \geq 5$ km, which would be consistent with the

observed GW characteristics at these altitudes presented by Baumgarten et al. (2018).

As mentioned in the introduction, the measurements of Baumgarten et al., (2017) show some evidence that the increase in the GWPED values with height is stronger during full-day- than nighttime by a factor of about 2, or, assuming roughly a 2:1 relation of daylight and nighttime (16 hours daylight and 8 hours nighttime) for summer high mid-latitudes, stronger during daylight than nighttime by a factor about2.5. For comparison, the estimated effect of ozone-temperature coupling for these

latitudes (factor 1.7) is somewhat smaller and would lead to an increase of the nighttime GWPED in the order of ~50% (0.7:1.5) of the observed increase. Conclusively, although the difference derived by Baumgarten et al., (2017) might be uncertain as mentioned in the introduction, and although the approach of the present paper cannot entirely explain the details of specific local measurements during a specific time period, the comparison confirms that ozone-gravity wave interaction might be able to produce significant daylight-nighttime differences in the GW amplitudes at high summer mid-latitudes.

Current state-of-the-art general circulation models (GCMs) usually use a variety of prescribed tropospheric sources and tuning parameters in the parameterized gravity wave drag (GWD) parameterizations forcing the middle atmospheric circulation (e.g., McLandress et al., 1998; Fritts and Alexander, 2003; Garcia et al., 2017), where the extreme low temperatures observed in the summer upper mesosphere provide an important benchmark for the quality of the upwelling branch and the associated adiabatic cooling produced by the models. Including ozone-gravity wave interaction into the

GCMs might lead to a substantial improvement of the used GWDs and the associated processes driving the summer mesospheric circulation, because the related increase in the GWPED must lead to a similar increase in the vertical momentum flux term determining the GWD. However, the incorporation of ozone-gravity wave interaction in a state-of-the-

art GCM using a GWD, or in a numerical model with resolved GWs, needs extensive test simulations, which is beyond the scope of the present paper.

Current GCMs particularly indicate significant changes in the time-mean circulation of the upper mesosphere due to the stratospheric ozone loss over Antarctica during southern spring and early summer via the induced changes in the GWD (Smith et al., 2010; Lossow et al., 2012; Lubi et al., 2016). Long-term changes in upper stratospheric ozone of up to $-8\%$ per decade derived from satellite measurements (e.g., Sofieva et al., 2017; WMO, 2018) could also affect the mesospheric circulation in the stratosphere and mesosphere by modulating the GW amplitudes and, hence, the GWD. Based on the

idealized approach of the present paper, we estimate the sensitivity of the amplification of the GW amplitudes in the upper mesosphere on changes in the ozone background $\mu_0$ and the ozone-related heating rate $Q_0(\mu_0)$, revealing that, for horizontal and vertical wavelengths $L_k \geq 500$ km and $L_m \leq 5$ km, a change of $\pm 10\%$ in $\mu_0$ or $Q_0$ results in a change of $\pm 10\%$ to $\pm 20\%$ in the upper mesospheric GWPED. Conclusively, the summer mesospheric upwelling might be much more sensitive to the long-term changes in upper stratospheric ozone as has been suggested by the GCMs up to now.

In the approach of the present paper, the variations due to the diurnal cycle in stratospheric ozone and atmospheric tides are excluded to examine the potential effect of ozone-gravity wave interaction as clear as possible based on standard equations describing upward propagating GWs in a constant background. On the one side, these variations can principally modulate the effect of ozone-gravity wave coupling by changing the planetary-scale background conditions for the propagation of the mesoscale GWs. Assuming – to a first order – linear modulations in the background ozone and background lapse rate

according to observed diurnal or tidal variations, the sensitivity calculations of the present paper suggest that the related modulations in the amplitude amplification are smaller than the effect of ozone-gravity wave coupling by approximately one order. Further test calculations have shown that the use of a height-dependent scale height $H(T_0)$ instead of a constant scale height $H_0$ at the levels of relevant amplification leads to stronger amplitude amplifications particularly over the summer south pole, because of the high temperatures in the stratopause region and the very low temperatures in the upper

mesosphere, where the related differences are also smaller than the first-order process (e.g., in the GWPED, for vertical wavelengths between $L_m = 5$ km and $L_m = 3$ km, between about 7.5% to 20% at summer polar latitudes and less than 5% at summer mid-latitudes).

On the other side, short-term fluctuations in the balanced zonal and meridional winds due to atmospheric tides can principally lead to changes in the upward GW propagation characteristics, and, hence, to significant daylight-nighttime

differences in the growth of the GW amplitudes with height, including nonlinear feedbacks of the propagating mesoscale GWs to the short-term balanced flow components. Further, multistep vertical coupling processes producing secondary GWs in the mesosphere could depend on daylight-nighttime conditions or tidal variations, which could also produce significant daylight-nighttime differences in the growth of the GW amplitudes with height. Considering the remarkable strong effect of ozone-gravity wave coupling suggested by the present paper, we may speculate that it significantly affects these possible

changes in the GW amplitudes due to short-term fluctuations in the balanced winds or multistep vertical coupling. However, an unequivocal quantification of the effects of these processes and the involved nonlinear interactions on the daylight-

nighttime differences in the GWPED needs much more investigations, e.g., based on extensive GW resolving model simulations with interactive ozone photochemistry, which is beyond the scope of the present paper.

The results of the present paper might stimulate further daytime-nighttime observations of GW activity particularly at specific measurement sites where the GWs are usually characterized by specific horizontal and vertical wavelengths, e.g., downwind of specific mountain ridges (east of Rocky Mountains, Southern Andes or Norwegian Caledonides), which could be helpful to better understand of how ozone-gravity wave coupling is operating in situ.

**Data Availability**

Background data and programs visualizing the presented analytic solutions are available at the IAP archive under ftp://ftp.iap-kborn.de/data-in-publications/Gabriel/ACP2021/.

**Acknowledgment.**

The author thanks Hauke Schmidt (MPI-Met, Hamburg) for providing HAMMONIA background data. Thanks also to two anonymous reviewers for critical comments.

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

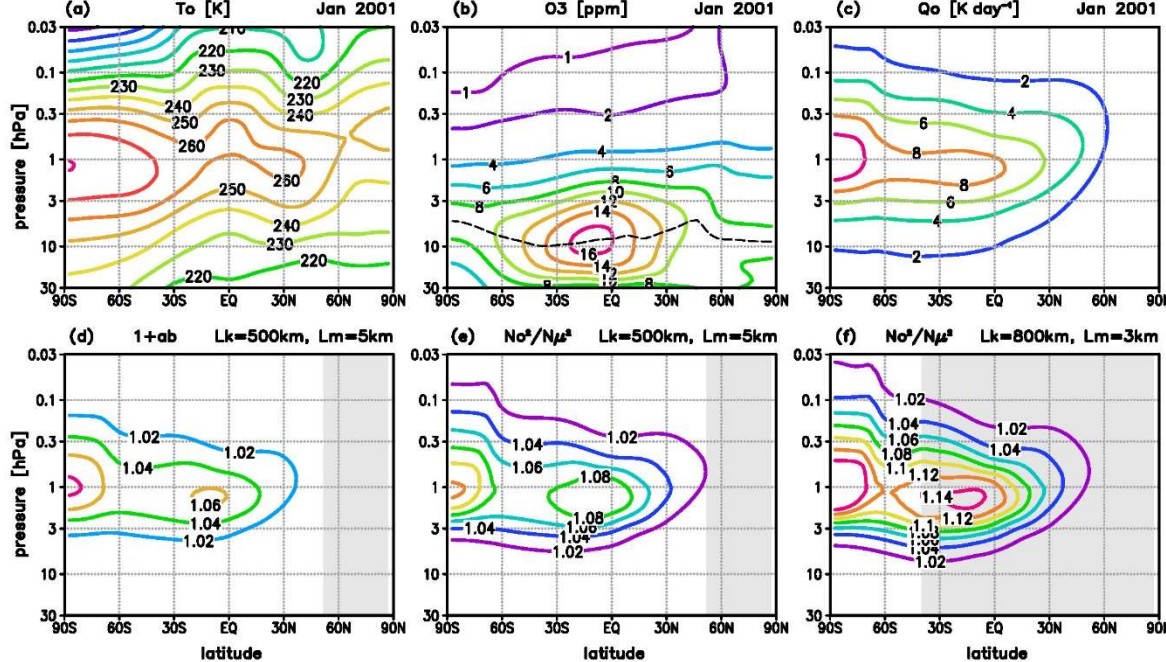

**Figure 1:** (a-c) Zonal and monthly mean background, (a) temperature $T_0$, (b) ozone mixing ratio $O_3$ (the dashed line denotes where $\partial O_3/\partial z=0$) and (c) ozone heating rate $Q_0$, January 2001, extracted from a simulation with the circulation and chemistry model HAMMONIA; (d-f) amplification factors (d) 1+ab and (e) $N_0^2/N_\mu^2$ for a GW with horizontal and vertical wavelengths $L_k$=500 km and $L_m$=5 km, and (f) $N_0^2/N_\mu^2$ for a GW with $L_k$=800 km and $L_m$=3 km; shaded areas denote the latitudes where the amplification is limited by the length of daylight ($\tau_i>\tau_{day}$).

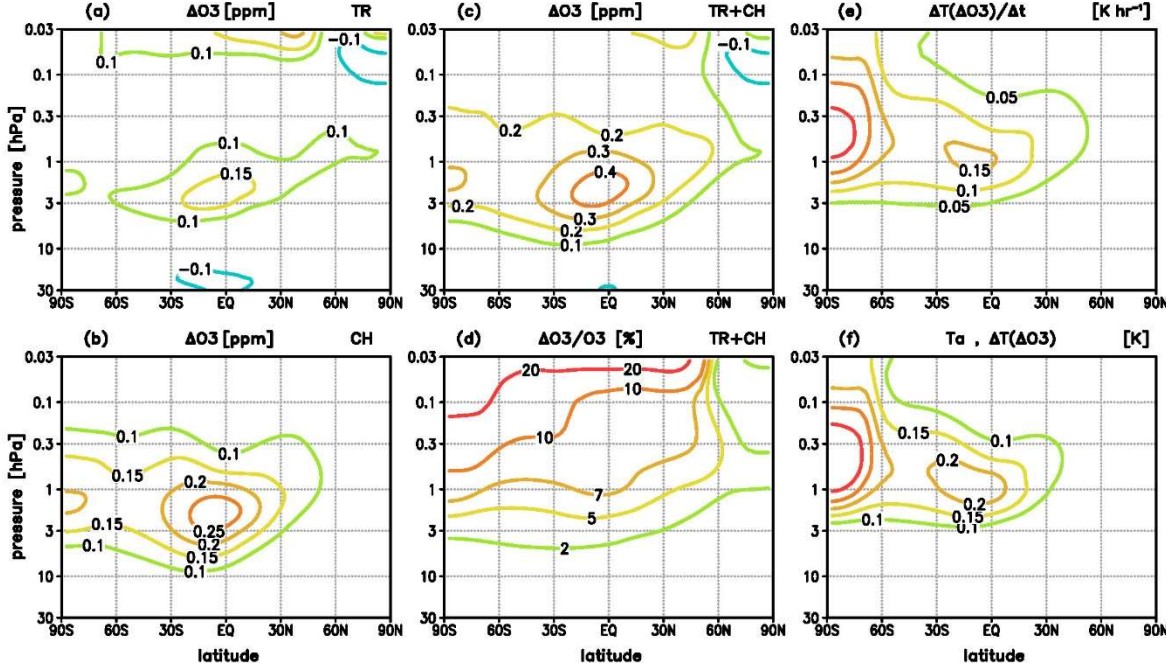

**Figure 2:** Changes due to ozone-temperature coupling at a specific level induced by an initial GW perturbation with horizontal and vertical wavelengths $L_k$=500 km and $L_m$=5 km, and with exponential increase in amplitude with height (initial temperature amplitude $T_a(zs)$=1 K at zs≈35 km (p=6.28 hPa)); (a) change in ozone due to vertical transport, (b) change in ozone due to photochemistry, (c) total change in ozone, (d) relative change in ozone, (e) change in the heating rate, (f) change in the temperature perturbation.

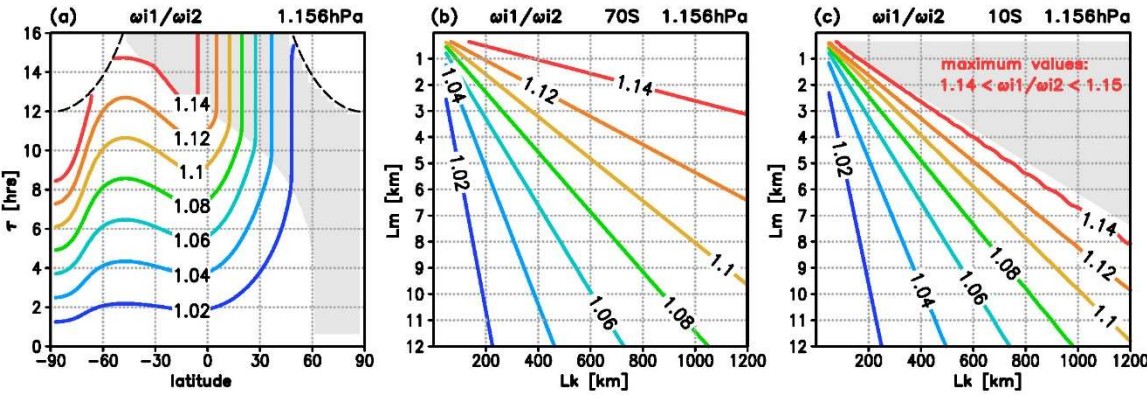

**Figure 3:** Amplification factor $\omega_{i1}/\omega_{i2}$ at a level of the maximum values of $\omega_{i1}/\omega_{i2}$ (1.156 hPa) illustrating the decrease of the intrinsic frequency with ($\omega_{i2}$) compared to without ($\omega_{i1}$) ozone-temperature coupling (compare with Figure 1e-f), (a) latitudinal distribution of $\omega_{i1}/\omega_{i2}$ as a function of the initial wave period $\tau_i$ [in hours], and (b-c) dependence of $\omega_{i1}/\omega_{i2}$ on the horizontal and vertical wavelengths $L_k$ and $L_m$ [in km] at (b) 70° S and (c) 10° S; shaded areas denote where the amplification is limited by the length of daylight ($\tau_i > \tau_{day}$).

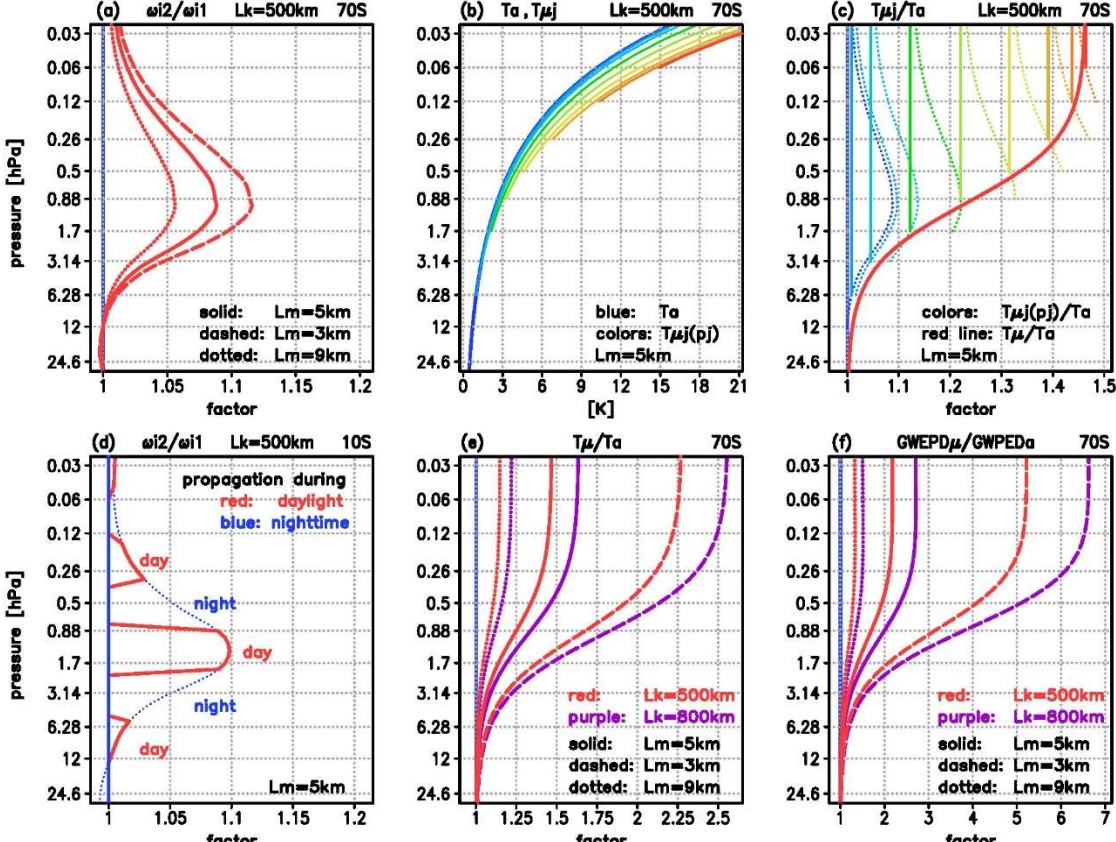

**Figure 4:** Illustration of the successive amplification of GW amplitudes during the upward level-by-level propagation, (a) amplification factor $\omega_{i1}/\omega_{i2}$ at 70° S for a GW with horizontal wavelength $L_k$=500 km and vertical wavelength $L_m$=5 km (red solid line), and, for comparison, $L_m$=3 km (dashed) and $L_m$=9 km (dotted); (b) temperature amplitudes for the GW with $L_k$=500 km and $L_m$=5 km, depicting the initial perturbation $T_a$ (blue) and the successively amplified amplitudes $T_{\mu j}(z_j)|_{j=1,n}$ (light blue towards red; here, n=8 for $L_m$=5 km), (c) same as (b) but for the relative amplitudes $T_{\mu j}(z_j)|_j/T_a$ (solid lines) together with the profiles of the previous level multiplied by $\omega_{i1}/\omega_{i2}$ (i.e., $T_{\mu j-1}(z_{j-1}) \cdot (\omega_{i1}/\omega_{i2})$, dashed lines) and a fitted approach $T_\mu$ (thick red solid line, defined by Eq. 18), (d) same as (a) for the case $L_k$=500 km and $L_m$=5 km but at 10° S including the limitation due to the length of night-time conditions, (e) relative values $T_\mu/T_a$ at 70° S for different horizontal (red: $L_k$=500 km, purple: $L_k$=800 km) and vertical (dashed: $L_m$=3 km, solid: $L_m$=5 km, dotted: $L_m$=9 km) wavelengths, (f) same as (e) but for the relative values $E_\mu/E_a$ of the related gravity wave potential energy density (GWPED, defined by Eq. 19).

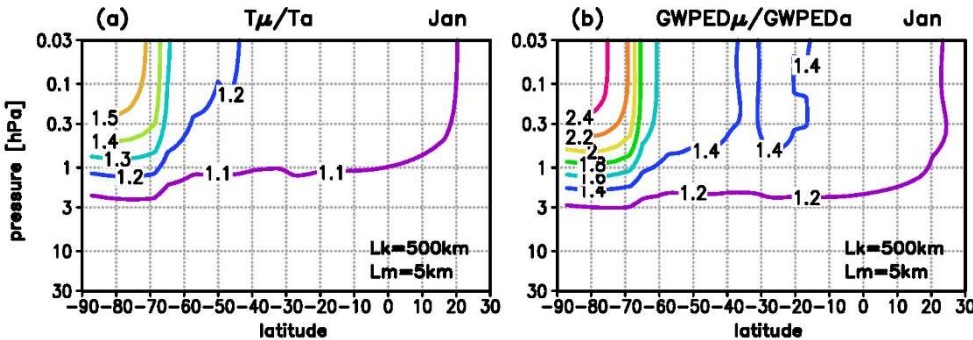

**Figure 5:** Cumulative amplification of the GW amplitude during the upward level-by-level propagation for a GW with $L_k$=500 km and $L_m$=5 km, (a) cumulative increase in the temperature amplitudes described by $T_\mu/T_a$, (b) related increase in the gravity wave potential energy density (GWPED) described by $E_\mu/E_a$; background conditions: January 2001.

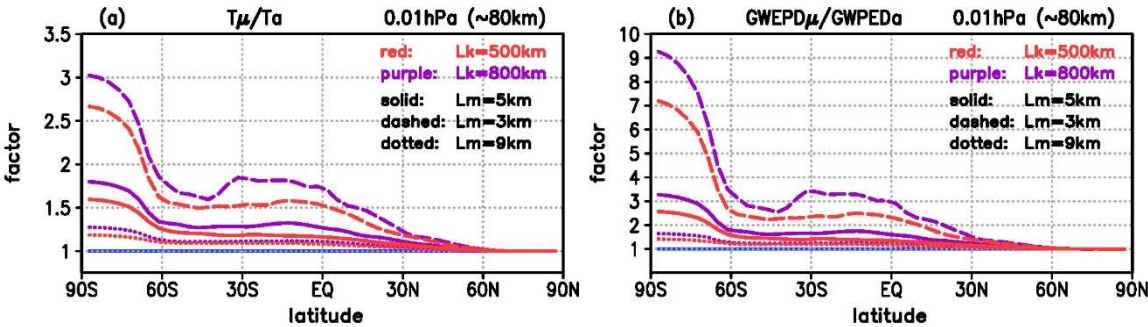

**Figure 6:** Cumulative amplification of the GW amplitudes similar as in Figure 5 but at upper mesospheric levels (0.01 hPa, ~80 km) for different horizontal and vertical wavelengths $L_k$ (red: 500 km, purple: 800 km) and $L_m$ (dotted: 9 km, solid: 5 km, dashed: 3 km), (a) $T_\mu/T_a$, (b) $E_\mu/E_a$.

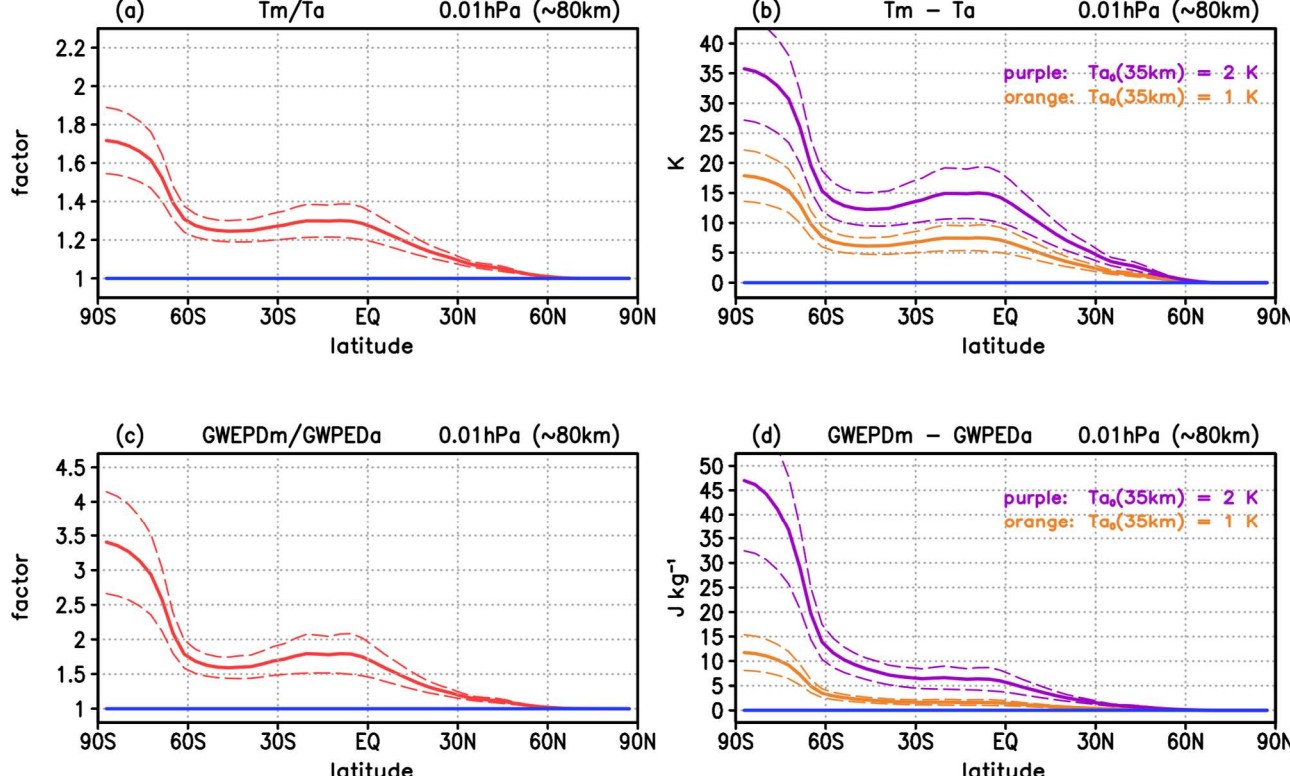

**Figure 7:** Similar to Figure 6, but for both relative and absolute changes of mean values averaged over 16 representative mesoscale GW events with different horizontal and vertical wavelengths $L_k$ and $L_m$ ($L_k$: 200, 500, 800 and 1100 km, $L_m$: 3, 5, 7 and 9 km), (a) relative change in temperature amplitude $T_m/T_a$ (red solid line; dashed lines denote the standard deviation), (b) absolute change $T_m-T_a$ for the case of an initial temperature perturbations $T_{a0}$ of 1 K (orange line) and 2 K (purple line) in the middle stratosphere, (c) and (d): same as (a) and (b) but for the GWPED (i.e., for $E_m/E_a$ and $E_m-E_a$).