# Peer review of "Ozone-Gravity Wave Interaction in the Upper Stratosphere/Lower Mesosphere"

_Atmospheric Chemistry and Physics, 2021_

## Author Comment (AC1)

Reply to the comments of Referee #1 on the manuscript acp-2021-1066

Thanks to Referee #1 for critical comments. First, I would like to give a general reply, and then a point-by-point response below (for orientation, the comments of Referee #1 are included in *Italic*).

General reply

Obviously, some points of the paper should be clearer from the beginning. The paper is indeed motivated by the conclusion of Baumgarten et al. (2017) that daytime-nighttime conditions have a significant but unexplained effect on the GWPED in the summer stratopause region; however, the paper does not intend to claim that all details of the observed GWPED profiles *can be solely explained* by ozone-gravity wave interaction as obviously concluded by Referee #1. The aim of the paper is to demonstrate that ozone-gravity wave interaction can principally lead to a significant amplification of gravity wave amplitudes, based on a theoretical approach excluding other perturbations like small-scale diffusion (l. 278), but also excluding other perturbations like atmospheric tides or so-called secondary gravity waves found in idealized model simulations, which are not explicitly discussed in the paper as required by Referee #1.

I agree that some revisions in the text and some more discussion could be helpful to make these points clearer. I also agree with Referee #1 that "*the theoretical model of dynamical coupling of the ozone heating rate with wave dynamics is certainly of interest*"; accordingly, the required points (critical level filtering due to the zonal wind, atmospheric tides, secondary gravity waves) will be included in the introduction and discussion of the revised manuscript.

*General Comment:*

*Gravity waves (GW) are a major source of the internal variability of the middle atmosphere. Motivated by lidar observations there is a claim that the gravity wave potential energy density (GWPED) during daylight can be enhanced compared to nighttime measurements at the upper stratosphere and mesosphere. This study seeks to present a theoretical approach to explain this enhancement by gravity wave-ozone interaction, due to changed heating/cooling rates caused by the vertical transport of air parcels by GW assuming idealized inertia gravity waves and an upward level-to-level propagation. The derived theoretical model of GW-ozone interaction was implemented in the well-established HAMMONIA model and all results are based on such model runs.*

I do not really understand the last sentence. Note here, for avoiding any misunderstandings, that the preprint uses HAMMONIA data only as a prescribed constant background for the analytic solutions derived in Section 2 (ll. 95-99); the analytic solutions describing ozone-gravity wave interaction (Sections 2.1 and 2.2) were not implemented in the HAMMONIA model, and related model runs were not carried out.

*However, there are major (almost fatally flawed) concerns to some parts of the submitted paper, which certainly require a more controversial and critical scientific analysis to support the results.*

As far as I understand, the *major (almost fatally flawed) concerns* of Referee #1 are related to the motivation of the paper by the full-day measurements of the GWPED published by Kaifler et al. (2015) and Baumgarten et al. (2017), and to a lack in discussing the seasonal cycle of the GWPED in relation to other relevant processes like critical layer filtering, atmospheric tides, and secondary gravity waves.

Perhaps it is meaningful to emphasize from the beginning that the preprint does not intend to analyze all the details of the observed GWPED profiles, or the reliability of these profiles which was already done in the cited papers. The preprint is motivated by the daytime-nighttime differences in the GWPED published by Baumgarten et al. (2017) revealing an unexplained stronger GW activity in the summer stratopause region during daytime than nighttime (compare Figure 6 and Figure 9 of Baumgarten et al., 2017). This must have an effect on the seasonal cycle at all latitudes, particularly on the change between polar night (or winter) and polar day (or summer), although – of course – not all features of the observed seasonal cycle can be explained by the daytime-nighttime differences. The aim of the preprint is to examine and to quantify the potential contribution of ozone-gravity wave interaction to this unexplained phenomenon of daytime-nighttime differences excluding any other processes contributing to the observations.

Related revisions in the introduction and discussion of the manuscript will be included following the replies to the specific comments of Referee #1 below.

Specific comments:

*While reading the manuscript, the reviewer usually browses the web is to collect background information. During this search, I noticed that the Institute of the Author listed a similar paper with the same title as accepted publication in ACP. If the paper is already accepted this review might already be obsolete (see attached screenshot from 31.01.2022).*

Thank you very much for this hint. This was a mistake of our administration managing the list of papers of the author's institute. I had ordered to put the citation of the ACPD preprint into the list as a possibility to stimulate the open discussion, but – of course – not an accepted ACP paper. I am very sorry that I didn't realize this mistake earlier. The citation is removed.

*Lidar observations have become a standard technique to measure temperature fluctuations in the middle atmosphere. Already a few decades ago such observations were used to derive GWPED. This study was motivated by lidar observations conducted during a campaign at the Davis station (69°S) in Antarctica (Kaifler et al., 2015) and mid-latitude observations at Kühlungsborn (54°N) (Baumgarten et al., 2017,2018). The reviewer did look at all three publications and tried to understand what is mentioned on page 3 lines 53-62. The Antarctic observations (Kaifler et al., 2015) are seasonal summer and winter differences and do not allow to distinguish a day-night comparison and, thus, it is hard to attribute the seasonal GWPED difference between the stratosphere and mesosphere to be caused by GW-ozone interaction. The seasonal differences of the tropospheric GW sources and mean circulation at the middle atmosphere should be considered and are likely contributing a lot to these differences. Secondly, the wind profile is dramatically different between a polar summer and winter condition, which directly affects the critical level filtering due to the strong zonal wind reversal at the summer MLT.*

(*Reply to comment on seasonal summer and winter differences*) The Antarctic observations illustrated in Figure 6 of Kaifler et al. (2015) show monthly means, therefore it allows to distinguish polar day conditions (December-January) from other months. Following Baumgarten et al. (2017), the GWPED at mid-latitudes is significantly stronger during daylight than nighttime (as summarized at ll. 56-61). It is evident that the related process must contribute to observed polar day-polar night differences (the question is how large), and therefore to the observed seasonal cycle or latitudinal dependence of the GWPED. This might become somewhat clearer as follows.

Critical level filtering occurs during strong westerlies between April and October (see Figure 7 of Kaifler et al., 2015), and the relative increase in GWPED with height (see Figure 6 of Kaifler et al., 2015) is stronger during polar day (factor ~6 during December and factor ~13 during January) than during times of increasing or decreasing nighttime without strong westerlies (factor ~5.3 during February and factor ~4.5 during November). Roughly estimated, the relative increase is stronger during polar day (mean values of December and January) than during February or November by a factor of about ~2, suggesting that the responsible process enhancing the GWPED becomes much stronger when changing from summer mid-latitudes to polar day conditions. This supports – from my point of view – the preliminary thesis that the stronger amplitudes during polar day could be related to the same process responsible for the daylight-nighttime differences found by Baumgarten et al. (2017); however, it does not intend to give a final explanation of the whole seasonal cycle of the observed GWPED but only an illustration of the gap in understanding the observations.

(*Reply to comment on seasonal differences in tropospheric GW sources*) Of course, the mesospheric GWPED depends on the GW sources in the troposphere or stratosphere; the question is what happens in between. In summary, the preprint provides better understanding of the principal dependence of the mesospheric GWPED on the stratospheric GW sources if ozone-gravity wave interaction is considered. This might be more clearer as follows.

GW sources are specified by GW amplitudes and other GW characteristics (horizontal and vertical wavelength, or intrinsic frequency). The seasonal differences of the amplitudes of the stratospheric GWPED amplitudes and the resulting mesospheric GWPED are – for example – given in Figure 6 of Kaifler et al. (2015) and Figures 6 and 9 of Baumgarten et al. (2017). The focus of the preprint is the relative ratio between the mesospheric and stratospheric GW amplitudes (which shows the discussed strong daylight-nighttime or polar day-polar night differences) in case of a specific prescribed GW source in the middle stratosphere, assuming exponential growth of the GW amplitude with height as first guess. This approach allows simplified solutions describing the principal process of ozone-gravity wave coupling for any initial GW perturbation excluding any other processes.

The preprint needs middle stratospheric GW sources as initial conditions. For evidence, the preprint assumes a moderate GW amplitude of 1 K in the middle stratosphere as starting point (page 8, line 245), which can be set to other values; however, these other values would not change the relative increase between middle stratosphere and mesosphere, either in case with or without ozone-temperature coupling. The dependence of the process on the other GW characteristics (horizontal and vertical wavelength, or intrinsic frequency) is discussed throughout the whole paper. Associating roughly the seasonal cycle and the latitudinal distribution of the GWPED in terms of daylight-nighttime conditions, Figures 5 and 6 of the preprint illustrate this effect of ozone-gravity wave interaction on the relative ratio between the mesospheric and stratospheric GWPED for different GW sources.

(*Reply to comment on critical level filtering*) Of course, the seasonal cycle in critical level filtering due to the zonal wind contributes to the seasonal cycle of the observed GWPED, as mentioned in the introduction of the preprint (page 3, lines 53-66). However, as discussed above, this process is operating during Apr-Oct when the westerlies are strong, but not during Nov-Mar, and the question arises why the GWPED increases much stronger during polar day than during Feb or Nov. Obviously there is a gap in understanding the observed GWPED during polar day, and the preprint examines a specific process to fill this gap.

For clarification, some revisions will be included in the introduction and discussion according to the replies above.

*At the mid-latitudes, Baumgarten et al., 2017 showed different climatologies of GWPED for different filtering methods. This points to another major concern when using the numbers. The GWPED seems to depend on the analysis method, which does not provide confidence that the ratios between the stratosphere and mesosphere can be derived reliable enough to support the hypothesis of the proposed GW-ozone effect. In particular, this is also mentioned in Kaifler et al., 2015 as well. Due to the decreased iron layer thickness during the summer at the MLT, the estimated GWPED values are more uncertain and sometimes not derivable applying the same filtering methodology. Erhard et al., 2015 also performed a detailed study to investigate the sensitivity of the different methods to estimate GWPED. These aspects deserve some more clarification in the introduction.*

The uncertainties of the cited Lidar measurements have not been discussed in the preprint because this is already explained and discussed in Baumgarten et al. (2017) and Kaifler et al. (2015), where both the stratospheric and mesospheric GWPED are shown as reliable results of these measurements. Baumgarten et al. (2017) explicitly discuss the daylight-nighttime differences in relation to the used filtering methods concluding that these differences might be of true geophysical origin (see Figures 6 and 9 of Baumgarten et al., 2017, and the related discussion), which is worthwhile enough for other researchers to examine a potential process leading to this phenomenon. However, for clarification, a related comment on the reliability of the measured GWPED will be included in the revised introduction.

*Another crucial concern when dealing with lidar and model data to investigate day-and-night differences are atmospheric tides. The ozone volume mixing ratio shows a very fast response to the terminator (sunlight) (e.g., https://doi.org/10.5194/acp-18-4113-2018). This time scale is much shorter than the investigated intrinsic gravity wave periods. Thus, it appears to be unlikely that an air parcel that is in the updraft part of an inertia gravity wave could sustain the volume mixing ratio over hours without getting back to the chemical equilibrium to the ambient atmosphere. Radiative processes seem to happen on much shorter time scales. Thus, the theoretical description of the paper might be correct, but the total effect could be much smaller as one needs a convolution with the time scales.*

Thanks a lot for the hint on the interesting paper of Schranz et al. (2018). However, I do not really understand the related critics on the present preprint.

In the upper stratosphere, ozone shows indeed a very fast response to changes in radiation because it is approximately in temperature-dependent photo-chemical equilibrium, which is

an essential preliminary of the preprint (ll. 67-68; ll. 146-164). Therefore, local ozone and temperature perturbations due to an upward propagating mesoscale gravity wave must nearly instantaneously lead to local changes in the temperature-dependent photo-chemical equilibrium, including nearly instantaneously coupled perturbations in ozone and temperature over the time-scale of the gravity wave perturbation compared to the unperturbed environment.

In comparison to the mesoscale gravity waves, atmospheric tides or the diurnal cycle of ozone are planetary-scale variations; indeed, they can change the background conditions for the local propagation of the mesoscale GW perturbations. One main point of the diurnal cycle is included in the preprint: the addressed process of ozone-gravity wave interaction is operating during daylight only but not during nighttime (this change is particularly important for slowly propagating GWs and considered in Section 2).

Following Schranz et al. (2018), the relative amplitude of the diurnal cycle of upper stratospheric ozone is in the order of 5% (summer solstice) to 8% (May); a somewhat stronger planetary-scale ozone background during daylight could principally lead to a somewhat stronger effect of ozone-gravity wave interaction on the mesoscale GW perturbations, analogously to a somewhat stronger or weaker effect in case of a long-term change in stratospheric ozone (in Section 2.2, ll. 391-399, the sensitivity is given in case of a change in the ozone background of 10%).

For clarification, the revised manuscript will include some more comments and discussion on the diurnal cycle of ozone. However, the theoretical approach calculates the effect of ozone-gravity wave interaction straight forward assuming a constant background and does not need any convolution of the time scales.

*Atmospheric tides are also important to estimate reliable GWPED. Baumgarten et al., 2019 (https://doi.org/10.5194/angeo-37-581-2019) demonstrated that there is also some interday tidal variability. Most of the above mention filtering techniques do not account for tides, which have almost similar or larger amplitudes compared to gravity waves at the stratosphere and mesosphere. Thus, the GWPED needs to be corrected for such tidal contaminations. This is also an issue for the HAMMONIA data, which is also affected by tidal modes. It remains unclear how day-night differences could be distinguished from the diurnal excitation due to the ozone absorption and associated heating rates. The advantage of tides is that the migrating tidal modes DW1, SW2, TW3 are sun-synchronous and fulfill the requirements assumed for the theoretical framework presented in the submitted manuscript. GW have random phases concerning their temporal behavior due to the various excitation mechanisms therefore it is unlikely that the updraft phase remains sun-synchronous, which is the key assumption in the manuscript. More likely is a random superposition of GW and a potential cancelation of the updraft and downdraft phases, which may result in a total zero effect.*

Yes, considering atmospheric tides is certainly important to understand the details of the observed GWPED variability, and it is challenging to separate mesoscale GW perturbations and planetary-scale tidal variations from a time series of measured GWPED at a specific location. However, this is not the purpose of the preprint. The theoretical approach of the preprint intends to examine the effect of ozone-gravity wave interaction on the increase of mesoscale gravity wave amplitudes with height in the upper stratosphere/lower mesosphere

region excluding – for a first guess – other perturbations or variability. However, an additional comment in the revised manuscript might be necessary that tidal variations can principally modulate the planetary-scale background for the propagation of mesoscale gravity waves, and that an amplification of GW amplitudes due to ozone-gravity wave interaction can principally lead to a stronger dissipation of the tidal signals in observations.

I do not really agree with the statement that tides *have almost similar or larger amplitudes compared to gravity waves at the stratosphere and mesosphere*. Particularly in the stratosphere and lower mesosphere, gravity wave amplitudes can be much larger than tidal variations. For example, the GW amplitudes forced by the Rocky Mountains can be one order larger (5 K to 10 K) than the tidal variations between 30 km and 70 km altitude shown in the above recommended paper of Baumgarten and Stober (2019).

Note also again that the preprint does not analyze any variations of the HAMMONIA data. The HAMMONIA data are only used for prescribing a constant monthly mean background for the theoretical approach of upward propagating gravity waves. Any diurnal variations or tides produced by the HAMMONIA model are not considered.

In this context I do not really understand the comment of *a potential cancelation of the updraft and downdraft phases, which may result in a total zero effect.* The gravity wave is a sinusoidal perturbation oscillating between (positive) updraft and (negative) downdraft, analogously to the swing of a pendulum, and the result of the preprint is that the amplitude is increasing while the frequency is decreasing (i.e., the time between the maximum and minimum of the oscillation becomes larger) when propagating through the upper stratosphere/lower mesosphere. Following Section 2, this process can be expressed in terms of the static stability in the dispersion relation. This process depends on the planetary-scale background conditions, which is varying – for example – due to long-term changes in ozone or changes in the static stability due to tidal variations; however, I do not see any cancelation of the updraft and downdraft phases.

Some related comments on atmospheric tides will be included in the revised manuscript to clarify this point from the beginning.

*The results indicate that the effect of gravity-wave-ozone coupling is most pronounced above the stratopause. Recently, a concept called multi-step vertical coupling (MSVC) was introduced Becker and Vadas, 2018 and later publications. Primary GW launched in the troposphere such as mountain waves, frontal waves, jet instabilities, etc. propagate vertically and dissipate generating a body force, which again causes secondary waves, which propagate further upward and so forth up to the thermosphere.*

Yes, these secondary waves found in idealized model simulations could be a significant factor contributing to the observed GWPED. However, the preprint wants to highlight the effect of ozone-gravity wave interaction on the amplitudes of upward propagating gravity waves, therefore any other perturbations like secondary gravity waves are excluded as a first guess. If the multi-step process producing secondary gravity waves described by Becker and Vadas (2018) is operating in the real atmosphere, the process of ozone-gravity wave interaction might have a significant effect on the amplitudes of these waves as far as they are vertically propagating through the upper stratosphere/lower mesosphere; however, this is speculative at the moment and needs some more numerical model simulations with both resolved gravity

waves and interactive ozone photochemistry, which is beyond the scope of the present preprint. A related comment will be included in the revised introduction and discussion.

*Considering the above-mentioned physical processes it appears to be unlikely that the ratios between the stratospheric and mesospheric GWPED can be solely explained by the proposed GW-ozone interaction. MSVC, the horizontal propagation of GW, or atmospheric tides play also important roles and deserve a detailed and critical assessment in this regard to understand the vertical profile of GWPED.*

As mentioned in the general reply above, the preprint does not want to claim that the observed GWPED profiles mentioned in the introduction *can be solely explained* by ozone-gravity wave interaction. The preprint also does not want to explain all details of the observed GWPED. The aim of the preprint is to demonstrate that ozone-gravity wave interaction can significantly contribute to the unexplained daylight-nighttime differences in the gravity wave amplitudes. Therefore, the theoretical approach excludes other perturbations or variability modulating the background, like secondary gravity waves or atmospheric tides, to highlight this process. Some more comments on the specific points of Referee #1 will be included in the revised introduction and discussion of the manuscript, as outlined above.

*However, the theoretical model of dynamical coupling of the ozone heating rate with wave dynamics is certainly of interest but should be contextualized with atmospheric tides and tidal excitations. The claim in the abstract that "ozone-gravity wave interaction is largely responsible for this effect" is certainly not so straightforward justified given the other dynamical aspects and the idealized model simulations.*

As mentioned above, I completely agree with Referee #1 that "*the theoretical model of dynamical coupling of the ozone heating rate with wave dynamics is certainly of interest*". In the abstract, a revised statement that "ozone-gravity wave interaction can significantly contribute to this phenomenon" (i.e., to the stronger increase in amplitudes with height during daylight than nighttime) might be better than "ozone-gravity wave interaction is largely responsible for this phenomenon".

A revised manuscript and an additional point-by-point reply to the comments of Referee #1 will be uploaded following the regulations of ACP.

---

## Author Comment (AC2)

Reply to the comments of Referee #2 on the manuscript acp-2021-1066

Thanks also to Referee #2 for critical comments. Please find a point-by-point response below (for orientation, the main comments of Referee #2 are numbered and included in *Italic*).

*The paper presents a possible mechanism of amplitude amplification of gravity waves by the interaction between ozone and gravity waves in the upper stratosphere/lower mesosphere. The paper is divided into three parts: an introduction, a section on the interaction between ozone and gravity waves, and a section titled "Summary and Conclusions." There are 6 Figures that present the results. I had difficulty following the content of the paper for several reasons.*

1) *First, the paper is written very compactly. The derivation of the main equations proving the positive feedback of the ozone-gravity wave coupling uses components from different sources, and I would have liked a clearer separation to make it easier for the reader. Also a clear distinction between methodology and results would be most welcome! Therefore, I propose to revise the layout of the manuscript and make it clearer.*

Yes, I agree, perhaps some more separation will be helpful for the reader. The revised manuscript will include some more subsections to separate the steps of the theoretical approach and the results. Section 2.1 will be separated into subsections including (a) the basic equations (ll. 107-143), (b) the derivation of the coupling parameters (ll. 144-193), (c) the formulation of the local amplification (ll. 194-235), and (d) the results illustrating the local amplification (ll. 236-277); Section 2.2 might be separated into subsections more related to the methodology (ll. 277-361) and presenting results (ll. 362-399).

2) *The second aspect may be a misunderstanding on my part: I cannot accept the dynamical concept of assumed gravity wave-ozone coupling (heating rate). My understanding is that propagating internal gravity waves cause positive and negative vertical displacements of the background airflow. Therefore, air transported through a gravity wave experiences both adiabatic cooling and heating. It seems to me (I found no other reference in the text) that only positive vertical velocities (i.e., displacements) are considered here to establish the "successive" or "cumulative amplitude amplification".*

Of course, a sinusoidal gravity wave perturbation includes both a positive (w'>0) and a negative (w'<0) component. The feedback of ozone-temperature coupling to the initial perturbation is described by an initial positive component of the updraft as an example (increase of w' if w'>0), but it is – of course – also valid for the accompanying negative component (decrease of w' if w'<0). Accordingly, as described and illustrated in Section 2.1, the amplitude (or the difference between the maximum and minimum of the oscillating wave pattern) is increasing while the frequency is decreasing (i.e., the time between the maximum and minimum is increasing) when propagating through the upper stratosphere/lower mesosphere. From my point of view this is easy to understand; however, the text of the manuscript will be revised to make this point clearer from the beginning, perhaps with an additional figure illustrating the changes in the oscillating structure of a gravity wave.

*Averaged over a horizontal wavelength or one period, the net effect of gravity wave-induced cooling and warming should be zero. In conclusion, I don't see any point in publishing the results as they have been written up now. A better presentation of the underlying concept is urgently needed. Again, I could be wrong: reading the text, I would assume that gravity wave-ozone coupling leads to an increase in background temperature when gravity waves are present and ozone photochemistry is working. Is this correct? I hope, I'm right in this aspect. If not, any clarification of the dynamical concept in the paper is highly appreciated.*

I hope that I understand these comments correctly, but I am not sure. Generally, an average over the horizontal wavelength or one period of a sinusoidal wave perturbation, either in case with or without ozone-temperature coupling, does not make sense, it must be zero because it is a sinusoidal wave perturbation. The theoretical approach describes an upward propagating gravity wave in a constant background atmosphere; therefore, in this specific standard approach, the gravity wave cannot change anything in the background.

The approach of Section 2 suggests that the GW amplitude becomes larger while the frequency is decreasing during the vertical propagation if ozone-temperature coupling is considered; however, the process only leads to a *net effect* on the time-mean temperature and circulation if the stronger increase in amplitude with height leads to a change in gravity wave breaking processes, which is discussed in Section 3 (ll. 440-459). In current GCMs, this process is described by the gravity wave drag (GWD) parameterization which describes the transformation of the potential gravity wave energy of upward propagating GWs into those gravity wave flux terms that deposit heat and drive the circulation in case of gravity wave breaking conditions. Including ozone-temperature coupling might lead to an improvement of the GWDs; however, a more detailed investigation needs extensive model calculations which are beyond the scope of the preprint. Some revisions in the text and some more comments will be included to make this point clearer.

3) *There is a third point that should be considered in a new version of the manuscript. The whole gravity wave concept relies on linear wave theory. However, the authors use a density scale height H that is strictly only applicable for an isothermal atmosphere as it is constant with altitude. Already in the textbooks by Gill (1982, page 50 top) and by Dutton (1976, pages 67-68) altitude-dependent scale heights are mentioned or proposed. Recently, Reichert et al. (2021) used a height-dependent H for investigating conservative growth rates from ground-based lidar measurements. So, it would be worthwhile to estimate the amplitude growth in an atmosphere with temperature varying with altitude. Especially, in the summer mesosphere where the temperatures can drop drastically from the stratopause to the cold mesopause, this effect might account for some of the observed exponential increase.*

Yes, the height-dependence of the density scale height H can have an effect on the amplitude growth especially in the summer mesosphere. The preprint focusses on the proposed effect of ozone-gravity wave coupling in the upper stratosphere/lower mesosphere, therefore the standard approach H=const might be suitable. A height-dependent H might particularly affect the approach of upward propagating GWs described in Section 2.2 (at ll. 286-312) but it might not be stronger than the feedback of the changing vertical group velocity to the amplitude growth described at ll. 382-390, because H does not change in the vertical by more than 10% to 20% between 30 km and 70 km. However, it might be indeed worthwhile to include an additional comment, or – perhaps – an additional figure, in the revised manuscript illustrating this effect.

**4)** *Last but not least, I see an essential difference in the gravity wave regimes of the upper stratosphere and lower mesosphere between summer and winter. This picture results from Figure 6 of Reichert et al (2021): it shows almost no seasonal variability of Ep in the layer 65 to 80 km altitude in contrast to the layers below. Thus, the mesosphere seems to be a region where gravity waves always exist almost independent from the local excitation at the place of the observations. Where these waves come from, if they are from primary or secondary or other sources, I don't know but they seem to be present all the time. In conclusion, the strong summer increase can probably also be explained by the reduced local excitation conditions, i.e. the strongly reduced Ep values at lower layers. Sure, this is for one location in the lee of the Andes but it is a convincing example. By the way, there is a further aspect not discussed in the paper: the superposition of gravity waves from different sources entering the observational volume horizontally and leading to enhanced Ep values as indicated by Reichert et al. (2021) as well.*

The paper of Reichert et al. (2021) is indeed interesting, and the addressed points are worthwhile for the discussion of the revised manuscript; however, in Figure 6 of this paper, I see a pronounced seasonal cycle in the layer 65 to 80 km, with around 31.4 Jkg$^{-1}$ during summer and 82.4 Jkg$^{-1}$ during winter, although it is evidently less pronounced than in the layers below. The suggested explanation ("*the strong summer increase can probably also be explained by the reduced local excitation conditions*") could indeed explain a fraction of the different relative relation between mesospheric and stratospheric sources; however, this is not less speculative than any other thesis, even because seasonal changes in this relative relation are obviously much stronger in the full-day measurements at the somewhat more southward located Davis (69°S) including polar day and polar night conditions (Kaifler et al., 2015, Figure 6).

However, please note here that the preprint does not want to explain all the details of measured stratospheric and mesospheric GW amplitudes or GWPED by the proposed effects of ozone-gravity wave interaction. The GWPED measurements are cited in the introduction as a motivation specifying the open questions and to motivate the purpose of the preprint. From my point of view, it is evident that an unexplained process responsible for daylight-nighttime differences in the GWPED, as found by Baumgarten et al. (2017, Figures 6 and 9), must have an influence on polar day-polar night differences, and therefore on the seasonal cycle, although – of course – it cannot explain all the details of the measured GWPED variability. For clarification, the introduction and the discussion of the revised manuscript will be somewhat rearranged, i.e., interpretations of the measured seasonal cycle will be strongly reduced in the abstract and in the introduction, whereas a discussion of the possible effect of ozone-gravity wave interaction on the seasonal cycle will be included in Section 4 (see also the reply to Referee #1).

*I would have liked to see the authors pay more attention to these possible dynamical aspects and their potential impact on growth rates. A discussion of both the dynamical and ozone temperature aspects would improve the paper and relate its new results to known published knowledge.*

As outlined above, the specific points of Referee #2 will be included in the revised manuscript. A revised manuscript with tracked changes and an additional point-by-point reply to the comments of Referee #2 will be uploaded following the regulations of ACP.

*Minor Comments:*

*line 48: "over-exponential" is probably not well-selected as term: what does it mean? I guess, you refer to exponential growth with a enhanced rate, correct?*

Yes, here "over-exponential" means that the exponential growth rate increases with height in difference to the usually assumed constant exponential growth rate. I think it is a usual expression; however, it is not really necessary and will be no longer used in the revised manuscript.

*line 79-80: here, the concept of w'>0 is introduced for the first time. I thought, well, why do the author not consider w'<0 as well as vertical displacements related to these vertical oscillations vary in time and space regularly in a gravity wave.*

Of course, the description is valid for both components of the wave (increase of w' if w'>0, decrease of w' if w'<0) suggesting an increase in the amplitude (or in the difference between the maximum and minimum of w') of the oscillating wave pattern. See also the reply to the main point 2) above.

*line 114: introduce minus sign in density equation*

Yes. Thank you.

*line 115: why is $v_0 \, d/dy$ missing in the total derivative?*

Yes, it is missing. This will be improved.

*line 238: Figure 8 of Reichert et al. (2021) shows that the majority of vertical wavelengths is about and large than 15 km. So, the choice of the selected parameters (especially with reference to the Andes) is not clear to me.*

The selected parameters are used as examples of GW characteristics where ozone-gravity wave interaction is particularly efficient, which becomes evident when discussing the dependence of this effect on the horizontal and vertical wavelengths (ll. 258-270). Indeed, a hint on possible sources (cyclones, Andes) is not necessary in this subsection and will be deleted; the relation to measured vertical wavelengths and possible sources are discussed in detail in Section 4 (ll. 430-439).

Perhaps one additional comment to the mentioned findings of Reichert et al. (2021). Based on idealized approaches, orographically forced GWs might have the same spatial scales as the smoothed mountain ridge (e.g., as a function of the half width and the maximum height if it is approximated by a Gaussian-type function); therefore, I would expect horizontal wavelengths of a few hundred km and vertical wavelengths of around 3 km to 5 km for GWs forced by the Andes. The measurement site (Rio Grande) of Reichert et al. (2021) is located at the southern end of South America and not really downwind of the Andes; therefore, the identified GWs at Rio Grande might be forced by cyclones or convective patterns travelling over the South Pacific, but not by the Andes, where the very large vertical wavelengths > 10 km might be more related to convection over the ocean (this is, of course, speculation and not issue of the preprint). Other papers suggest most pronounced vertical wavelengths between about 5 km and 9 km in the upper stratosphere/lower mesosphere region (e.g., Baumgarten et al., 2018).

However, the findings of Reichert et al. (2021) are interesting and will be included in the discussion of the vertical wavelengths (ll. 430-439).

*line 266: Why do you use "but" not "and"?*

Yes, thank you, "and" is right here.

*References:*

*Dutton, J. A., 1976: The Ceaseless Wind. 1st ed., McGraw-Hill, New York and London, 579 pp.*

*Gill, A. E., 1982: Atmosphere-Ocean Dynamics, Academic Press, 1st edn.,662 pp.*

*Reichert, R. et al. 2021: High-cadence lidar observations of middle atmospheric temperature and gravity waves at the Southern Andes hot spot. Journal of Geophysical Research: Atmospheres, 126, e2021JD034683. https://doi.org/10.1029/2021JD034683*

---

## Author Comment (AC3)

Reply to the comments RC2 of Referee #1 on the manuscript acp-2021-1066

Thanks again to Referee #1 for the comments to the first reply. Please find a second reply below (again, for orientation, the comments of Referee #1 are included in *Italic*).

General reply:

Obviously, the way how I introduce the motivation (which mirrors the way how it arose) leads to some serious misinterpretations. I follow the suggestion of Referee #1 that "*skipping the observations to a large extent and just presenting the results as an idealized theoretical approach that requires observational justification*" might be the most meaningful way to improve this point in a revised manuscript.

Statements in the text that could lead to such misinterpretations will be skipped or revised. The rough estimations based on the measured GWPED cited in the preprint and used for introducing the motivation will be skipped in both the abstract (l. 3) and the introduction (ll. 51-66). However, some introducing remarks might be allowed, e.g., the citation that Baumgarten et al (2017) "assumed an unexplained process of true geophysical origin responsible for the daylight-nighttime differences in the GWPED during summer months", including a hint on the uncertainty of such measurements.

In Section 2, the short statement that the relative increase in the GWPED of specific GWs calculated by the theoretical approach is "quantitatively in agreement with the observations" (ll. 428-429) will be deleted because it seems to be misleading. Instead, the relevance of the suggested process will be shortly discussed in Section 3 (Summary and Conclusions), in the context of other processes.

A justification of the relevance is given because – for specific GWs with large horizontal and small vertical wavelengths – the change in the relative increase between stratospheric and mesospheric GW amplitudes calculated by the theoretical approach is in the order of observed relative increases (this was thought to be the original message of ll. 428-429). If this change calculated by the theoretical approach would be much smaller, I would not have submitted the preprint because then the process might be not significant. However, as I tried to explain in the first reply, the preprint is a theoretical approach and does not want to *attribute the observed GWPED only to a specific gravity wave with defined properties.* From my point of view, it is evident that one single GW cannot explain an observed GWPED at a specific location. Moreover, at the current stage of research, it is generally not possible to attribute the observed GWPED only to one selected specific process like primary or secondary GWs or tides. However, based on the findings of the preprint it is allowed to conclude that ozone-gravity wave interaction might play a significant role in the middle atmosphere to stimulate further research works.

*Comment on public reply:*

*General Comment:*

*The reviewer appreciates the quick response to the raised concerns. However, the replies also caused further concerns on the manuscript and require clarification. The reviewer takes the freedom to rephrase the comments a bit to reduce the ambiguity.*

*HAMMONIA (minor comment):*

*In the acknowledgments, there is a statement about computational resources. If there were no computational resources used why acknowledge.*

Some few resources have been used for handling the data. However, perhaps this statement should be deleted to avoid misunderstandings.

*Day and night differences (major concern/ very critical):*

*The submitted paper points at Lidar observations conducted at the Antarctic and mid-latitudes. These measurements are essential to motivate the main narrative of the paper, but also to justify the results to be relevant. Thus, the paper should present a careful discussion of the observations in the context of this work. The shown GWPED in both publications includes all types of waves, viz. tides, planetary waves, and gravity waves. The different filtering approaches underline this aspect (Erhard et al., 2015, Baumgarten et al., 2017). There is a concern to generalize and attribute the observed GWPED only to a specific gravity wave with defined properties. The observational uncertainties are supposed to be mentioned and discussed here as well.*

Yes, I agree again: a variety of dynamic processes contribute to the observed GWPED at a specific location, and not anyone of them can solely explain these observations. In the revised abstract and introduction, the quantifying interpretations of observations will be skipped, and the motivation will be focused on those statements that can be find in cited publications. Some statements in the text will be deleted or revised to avoid the impression that the preprint wants to *attribute the observed GWPED only to a specific gravity wave with defined properties.* In the revised manuscript, the relevance of the suggested process will be discussed in Section 3 in the context of observations and other processes.

*There is another major concern when generalizing the polar day-night differences, which are, in fact, summer-winter seasonal differences and cannot be linked to the mid-latitude day-night difference. These are entirely different physical aspects due to critical level filtering, source variability, and gravity wave propagations conditions.*

*Looking at Figures 2 and 3 in Baumgarten et al., 2017 does not indicate any local time dependence of the gravity wave activity. Only monthly averaged GWPED results show a day-night difference. Baumgarten et al., 2017 even discussed the day-night differences as part of the analysis bias concerning tides. This was later confirmed by Baumgarten et al., 2019 when the day-to-day variability was analyzed combining spatial and temporal filters into one multi-dimensional retrieval. This is also an aspect for planetary waves and lidar observation as demonstrated by Eixmann et al., 2020 (AG), which is relevant for summer–winter comparison at the Arctic/Antarctic.*

Yes, also here I agree again, all details of the observed *summer-winter seasonal differences cannot be linked to the mid-latitude day-night difference.* The processes responsible for the

daylight-nighttime differences could play a role for polar-day-polar night differences, but much more research is needed to attribute the observed GWPED unequivocally to the different processes operating in the real atmosphere. The abovementioned aspects of Referee #1 can be included in the discussion in Section 3.

*Critical level filtering:*

*The reviewer strongly disagrees with the statement in the replies that "Critical level filtering occurs during strong westerlies between April and October (see Figure 7 of Kaifler et al., 2015)". Critical level filtering is present at all times and during all seasons, however, depending on the sign of the stratospheric winds different gravity waves encounter the critical level depending on their propagation direction and phase speed. This is directly related to the source questions and multi-step-vertical coupling processes.*

I do not really understand the strong disagreement. The first reply was clearly related to the change from westerlies to easterlies, which is undoubtedly the most important factor in the seasonal cycle of critical level filtering. Of course, critical level filtering is present during summer months; however, other processes influencing upward propagating GWs could be much more effective during summer because of the seasonal change in the zonal winds. Indeed, this change might be one of the most important factors leading to somewhat stronger mesospheric GWPED during summer than winter although the tropospheric GW sources are generally weaker during summer than winter.

However, in the revised manuscript, a short discussion of the potential relevance of ozone-gravity wave interaction will be given in Section 3, in the context of other relevant processes like critical level filtering.

*Tidal amplitudes and gravity wave amplitudes:*

*Tidal amplitudes (semidiurnal or diurnal) can reach up to 8-15K (stratosphere/lower mesosphere) and occasionally 20 K (mesosphere) at the middle atmosphere between 30-80 km (e.g., from MERRA2). However, the amplitudes of tides are altitude-dependent and undergo the same exponential growth as gravity waves. The reviewer does not agree and has not seen observational evidence for an order of magnitude difference between tidal and gravity wave amplitudes (in a statistical sense) at the stratosphere and mesosphere. None of the lidar observations that are presented in the motivation are even close to the Rocky Mountains.*

In the first response to the preprint, the reviewer #1 gave a general statement that tides *have almost similar or larger amplitudes compared to gravity waves at the stratosphere and mesosphere*, together with the hint on Baumgarten and Stober (2019) who found amplitudes of tides in the upper stratosphere/lower mesosphere in the order of 1 K to 2K, which is not almost larger than GW amplitudes. The example of the Rocky Mountains was only used as one counterexample to the general statement.

However, as above, the relevance of the suggested process of ozone-gravity wave interaction will be shortly discussed in Section 3, in the context of other processes like tides.

*Sinusoidal approximation of gravity wave:*

*The reply draws an analogy between a gravity wave and a pendulum. This approximation seems to be by far too idealized as it skips key properties of a wave for real atmosphere application as they are found in observations. Gravity waves have a 3-dimensional wave vector and an intrinsic period and often occur not as an isolated plane wave but in wave packages. These packages have an envelope function, which is often assumed/approximated to be Gaussian. Depending on the background flow and the properties of the wave trains in the package cancelation effects are likely as updraft and downdraft phases can mix for a fixed observer on the ground in the Eulerian frame of reference. A pure vertical 1 D approximation is fine as a theoretical approach, but hard to be generalized in a real environment.*

Of course, a gravity wave is usually described by an amplitude, a 3-dimensional vector, and an intrinsic period, as also used in the preprint (l. 132, l. 212) to derive the dispersion relations. The simple image of a pendulum was only used in the first reply to clarify that the discussed effect of ozone-temperature coupling amplifies both the maximum and minimum of the oscillating wave pattern, but it does not play any role in the preprint. The preprint uses a variety of single waves to highlight the process of ozone-temperature coupling as clear as possible. In summary, yes, the preprint is based on an idealized approach; however, the results might stimulate further research works based on observations or model simulations.

*In summary:*

*The reviewer values the theoretical approach presented in the manuscript but has serious concerns about the motivation and justification of its importance. A revision of this manuscript either requires dealing with all the observations in more detail, including atmospheric tides and other dynamical effects as well as their biases, or skipping the observations to a large extent and just presenting the results as an idealized theoretical approach that requires observational justification. The way how the amplitude growth is well-founded between theory and observations is not appropriate. However, a justification could be also achieved by performing ICON model runs with high resolution to investigate the presented approach with resolved gravity waves in more detail. In principle, this is also possible with HIAMCM. Gravity wave resolving models permit a less ambiguous wave characterization. Such model runs will certainly strengthen the presented conclusions if confirmed. However, the reviewer understands that the model runs are a lot of work and might be postponed to future work.*

Yes, I agree, abstract, introduction and discussion in Section 3 will be revised as outlined in the general reply above.

Indeed, currently I am going to perform UA-ICON model simulations with high resolution and interactively coupled ozone chemistry (with the help of some collaborators) for justifying the proposed process. This will need some time and is beyond the scope of the present preprint. However, it would be great if the results of the preprint would stimulate also other modelling groups to examine this process.

---

## Author Response (AR1)

Reply to the comments RC1 and RC3 of Referee #1 on the manuscript acp-2021-1066

Thanks again to Referee #1 for critical comments. In addition to the first and second reply in the open discussion, please find here the final reply to all comments including the specified changes in the revised manuscript (for orientation, the comments of Referee #1 are included in *Italic*).

General reply (summary)

As general statement to the major concerns: the paper does not want to claim that observed GWPED profiles *can be solely explained* by ozone-gravity wave interaction, or to *attribute the observed GWPED only to a specific gravity wave with defined properties.* This was, of course, not the intension of the paper. The aim of the paper is to demonstrate that ozone-gravity wave coupling can principally lead to significant amplitude amplifications and daylight-nighttime differences within an important range of mesoscale GWs, but not to provide a complete explanation of published GWPED profiles derived from measurements.

Accordingly, the motivation in the abstract and introduction is revised to avoid such a misunderstanding. Possible effects of ozone-gravity wave coupling on daylight-nighttime differences and polar day-polar night differences are now discussed in Section 3, including additional comments on other relevant processes (critical layer filtering by the zonal wind, atmospheric tides, secondary gravity waves). Additionally, section 2 (subsection 2.2.4) includes now information on the sensitivity of the effect of ozone-gravity wave coupling to the modulations of the background by the diurnal cycle of ozone or atmospheric tides.

*General Comment* (RC1)*:*

*Gravity waves (GW) are a major source of the internal variability of the middle atmosphere. Motivated by lidar observations there is a claim that the gravity wave potential energy density (GWPED) during daylight can be enhanced compared to nighttime measurements at the upper stratosphere and mesosphere. This study seeks to present a theoretical approach to explain this enhancement by gravity wave-ozone interaction, due to changed heating/cooling rates caused by the vertical transport of air parcels by GW assuming idealized inertia gravity waves and an upward level-to-level propagation. The derived theoretical model of GW-ozone interaction was implemented in the well-established HAMMONIA model and all results are based on such model runs.*

As mentioned in the first reply, there might be a misunderstanding: HAMMONIA data are only used as prescribed constant background; the analytic solutions describing ozone-gravity wave coupling were not implemented in the HAMMONIA model, and related model runs were not carried out (previous preprint: ll. 96-99; revised manuscript: ll. 92-95).

*However, there are major (almost fatally flawed) concerns to some parts of the submitted paper, which certainly require a more controversial and critical scientific analysis to support the results.*

*Specific comments* (RC1)*:*

*While reading the manuscript, the reviewer usually browses the web is to collect background information. During this search, I noticed that the Institute of the Author listed a similar paper with the same title as accepted publication in ACP. If the paper is already accepted this review might already be obsolete (see attached screenshot from 31.01.2022).*

Thank you again. As mentioned in the first reply, this was a mistake of our administration. The citation was removed after I received the reviewer's comment.

*Lidar observations have become a standard technique to measure temperature fluctuations in the middle atmosphere. Already a few decades ago such observations were used to derive GWPED. This study was motivated by lidar observations conducted during a campaign at the Davis station (69°S) in Antarctica (Kaifler et al., 2015) and mid-latitude observations at Kühlungsborn (54°N) (Baumgarten et al., 2017,2018). The reviewer did look at all three publications and tried to understand what is mentioned on page 3 lines 53-62. The Antarctic observations (Kaifler et al., 2015) are seasonal summer and winter differences and do not allow to distinguish a day-night comparison and, thus, it is hard to attribute the seasonal GWPED difference between the stratosphere and mesosphere to be caused by GW-ozone interaction. The seasonal differences of the tropospheric GW sources and mean circulation at the middle atmosphere should be considered and are likely contributing a lot to these differences. Secondly, the wind profile is dramatically different between a polar summer and winter condition, which directly affects the critical level filtering due to the strong zonal wind reversal at the summer MLT.*

This part of the introduction, which summarizes the motivation of the paper, is revised; the notes on the seasonal cycle are skipped; instead, it is only highlighted that the published GWPED measurements of Baumgarten et al. (2017) might be uncertain but interesting enough to stimulate the examination of the present paper (page 3, ll. 51-62).

The relevance of ozone-gravity wave coupling in relation to the GWPED values derived by Baumgarten et al. (2017, 2018) and Kaifler et al. (2015) is now discussed in Section 3, including a statement that the latter did not separate daylight-nighttime differences explicitly (Section 3, ll. 489-530, particularly ll. 512-514).

As a conclusion, the potential relevance of ozone-gravity wave coupling is highlighted; however, it might be evident that the paper does not want to attribute the seasonal differences in the GWPED solely to this process (ll. 39-40, ll. 500-503, ll. 526-530).

Also, to avoid such a misunderstanding, two short conclusions on the quantitative agreement between the effect of ozone-gravity wave coupling and the observed relative increase in the GWPED between stratosphere and mesosphere are deleted; instead, the results of the theoretical approach are just summarized before discussing the relevance in section 3 (previous preprint: ll. 373-375, ll. 428-429; revised manuscript: ll. 391-394, ll. 493-494 and ll. 504-505).

A note on the seasonal differences in the stratospheric GW sources is included, and the possible contribution to the mesospheric GWPED is discussed (ll. 510-511, ll. 520-526).

Of course, critical level filtering by the zonal wind plays an important role in the seasonal cycle of the GWPED; it is now highlighted in the discussion (ll. 515-516).

*At the mid-latitudes, Baumgarten et al., 2017 showed different climatologies of GWPED for different filtering methods. This points to another major concern when using the numbers. The GWPED seems to depend on the analysis method, which does not provide confidence that the ratios between the stratosphere and mesosphere can be derived reliable enough to support the hypothesis of the proposed GW-ozone effect. In particular, this is also mentioned in Kaifler et al., 2015 as well. Due to the decreased iron layer thickness during the summer at the MLT, the estimated GWPED values are more uncertain and sometimes not derivable applying the same filtering methodology. Erhard et al., 2015 also performed a detailed study to investigate the sensitivity of the different methods to estimate GWPED. These aspects deserve some more clarification in the introduction.*

The general uncertainties of the cited GWPED measurements and its daylight-nighttime differences are now explicitly mentioned in the introduction, including more explicitly the conclusion of Baumgarten et al. (2017) that the daylight-nighttime differences might be of true geophysical origin (ll. 51-62).

*Another crucial concern when dealing with lidar and model data to investigate day-and-night differences are atmospheric tides. The ozone volume mixing ratio shows a very fast response to the terminator (sunlight) (e.g., https://doi.org/10.5194/acp-18-4113-2018). This time scale is much shorter than the investigated intrinsic gravity wave periods. Thus, it appears to be unlikely that an air parcel that is in the updraft part of an inertia gravity wave could sustain the volume mixing ratio over hours without getting back to the chemical equilibrium to the ambient atmosphere. Radiative processes seem to happen on much shorter time scales. Thus, the theoretical description of the paper might be correct, but the total effect could be much smaller as one needs a convolution with the time scales.*

As stated in the first reply, I do not really understand all these critical points; obviously there was a misunderstanding concerning the constant background which does not include tidal variations (see above); however, I try to give a reply as far as I understand.

In the paper, nearly instantaneous temperature-dependent photo-chemical equilibrium is considered as an essential preliminary of the examination; accordingly, local ozone and temperature perturbations due to an upward propagating mesoscale gravity wave must nearly instantaneously lead to local changes in the temperature-dependent photo-chemical equilibrium, including nearly instantaneously coupled perturbations in ozone and temperature over the time-scale of the gravity wave perturbation compared to the unperturbed environment. The theoretical approach calculates this effect of ozone-gravity wave interaction straight forward assuming a constant background and does not need any convolution of the time scales (photo-chemical equilibrium is introduced in the introduction at ll. 63-64, and in subsection 2.1.2, ll. 145-165).

An interesting point could be indeed that atmospheric tides or the diurnal cycle of ozone are planetary-scale variations which can change the background conditions for the local propagation of the mesoscale GW perturbations. Based on the diurnal cycle of stratospheric ozone presented by the recommended paper of Schranz et al. (ACP, 2018), the revised manuscript includes an estimation of this effect in subsection 2.2.4 and a related note in the discussion (ll. 423-429, ll. 551-554).

*Atmospheric tides are also important to estimate reliable GWPED. Baumgarten et al., 2019 (https://doi.org/10.5194/angeo-37-581-2019) demonstrated that there is also some interday tidal variability. Most of the above mention filtering techniques do not account for tides, which have almost similar or larger amplitudes compared to gravity waves at the stratosphere and mesosphere. Thus, the GWPED needs to be corrected for such tidal contaminations. This is also an issue for the HAMMONIA data, which is also affected by tidal modes. It remains unclear how day-night differences could be distinguished from the diurnal excitation due to the ozone absorption and associated heating rates. The advantage of tides is that the migrating tidal modes DW1, SW2, TW3 are sun-synchronous and fulfill the requirements assumed for the theoretical framework presented in the submitted manuscript. GW have random phases concerning their temporal behavior due to the various excitation mechanisms therefore it is unlikely that the updraft phase remains sun-synchronous, which is the key assumption in the manuscript. More likely is a random superposition of GW and a potential cancelation of the updraft and downdraft phases, which may result in a total zero effect.*

The general problem of tides in estimating reliable GWPED values based on local time series is now discussed in the introduction, together with a note in section 3 (ll. 53-59, ll. 517-520).

However, again, there might be a misunderstanding. The paper uses monthly means of HAMMONIA as constant background, and any tides simulated by HAMMONIA are not considered; in the theoretical approach, atmospheric tides are excluded as a first guess, like other processes, as now explicitly mentioned (ll. 286-287).

In this context, there is also not any potential cancelation of the updraft and downdraft phases; the basic idea and the theoretical approach describes the amplification of both the ascent (wave crest) and descent (wave trough) of a sinusoidal GW pattern during its propagation through the USLM; for clarification, the text is improved from the beginning (abstract and introduction: ll., 32-34, ll. 73-83; section 2.1.2: ll. 142-146, ll. 163-165, ll. 185-187; section 2.1.4, ll. 253-254, ll. 262-264; section 3, ll. 462-464).

In addition, like for the diurnal cycle of ozone, the revised manuscript now includes an estimation and discussion of the potential effect of tides on the cumulative amplification of the GW amplitudes by modulating the background temperature, based on the tidal amplitudes between 30 km and 70 km altitude shown in the recommended paper of Baumgarten and Stober (2019), but also assuming much larger modulations; overall, the related effects are smaller than the first-order effect of ozone-gravity wave coupling by approximately one order (ll. 430-438, ll. 551-554).

*The results indicate that the effect of gravity-wave-ozone coupling is most pronounced above the stratopause. Recently, a concept called multi-step vertical coupling (MSVC) was introduced Becker and Vadas, 2018 and later publications. Primary GW launched in the troposphere such as mountain waves, frontal waves, jet instabilities, etc. propagate vertically and dissipate generating a body force, which again causes secondary waves, which propagate further upward and so forth up to the thermosphere.*

Secondary gravity waves might be an interesting phenomenon and a note is included in the discussion; however, in the theoretical approach, they are excluded as a first guess like other processes, as now explicitly mentioned (ll. 286-287; ll. 518-520).

*Considering the above-mentioned physical processes it appears to be unlikely that the ratios between the stratospheric and mesospheric GWPED can be solely explained by the proposed GW-ozone interaction. MSVC, the horizontal propagation of GW, or atmospheric tides play also important roles and deserve a detailed and critical assessment in this regard to understand the vertical profile of GWPED.*

*However, the theoretical model of dynamical coupling of the ozone heating rate with wave dynamics is certainly of interest but should be contextualized with atmospheric tides and tidal excitations. The claim in the abstract that "ozone-gravity wave interaction is largely responsible for this effect" is certainly not so straightforward justified given the other dynamical aspects and the idealized model simulations.*

As mentioned in the general reply above, the preprint does not want to claim that observed GWPED profiles *can be solely explained* by ozone-gravity wave interaction. The abstract, introduction and discussion of the manuscript are improved to avoid such a misunderstanding, including additional sensitivity calculations and discussion in relation to the other processes mentioned by the reviewer (abstract: ll. 28-30 and ll. 39-40, introduction: ll. 51-62, subsection 2.2.4: ll.423-438, section 3: ll. 493-530 and ll. 551-554).

*Comment on public reply:*

*General Comment* (RC3)*:*

*The reviewer appreciates the quick response to the raised concerns. However, the replies also caused further concerns on the manuscript and require clarification. The reviewer takes the freedom to rephrase the comments a bit to reduce the ambiguity.*

*HAMMONIA (minor comment):*

*In the acknowledgments, there is a statement about computational resources. If there were no computational resources used why acknowledge.*

As mentioned in the first reply, some few resources have been used for handling the data. However, this statement is deleted to avoid misunderstandings (l. 570).

*Day and night differences (major concern/ very critical):*

*The submitted paper points at Lidar observations conducted at the Antarctic and mid-latitudes. These measurements are essential to motivate the main narrative of the paper, but also to justify the results to be relevant. Thus, the paper should present a careful discussion of the observations in the context of this work. The shown GWPED in both publications includes all types of waves, viz. tides, planetary waves, and gravity waves. The different filtering approaches underline this aspect (Erhard et al., 2015, Baumgarten et al., 2017). There is a concern to generalize and attribute the observed GWPED only to a specific gravity wave with*

*defined properties. The observational uncertainties are supposed to be mentioned and discussed here as well.*

As described above, the related motivation in the introduction is improved, including statements on the uncertainties in relation to the filtering methods; the discussion in section 3 is extended in relation to other important processes contributing to local GWPED profiles derived from measurements (ll. 51-62, ll. 493-530).

As mentioned above, the paper does not want to *attribute the observed GWPED only to a specific gravity wave with defined properties.* The results show a significant effect of ozone-gravity wave coupling within a wide range of mesoscale GWs, therefore it might be relevant for the middle atmospheric circulation. For clarification, the text of the manuscript is improved (ll. 39-40, ll. 493-504, ll. 526-530).

*There is another major concern when generalizing the polar day-night differences, which are, in fact, summer-winter seasonal differences and cannot be linked to the mid-latitude day-night difference. These are entirely different physical aspects due to critical level filtering, source variability, and gravity wave propagations conditions.*

*Looking at Figures 2 and 3 in Baumgarten et al., 2017 does not indicate any local time dependence of the gravity wave activity. Only monthly averaged GWPED results show a day-night difference. Baumgarten et al., 2017 even discussed the day-night differences as part of the analysis bias concerning tides. This was later confirmed by Baumgarten et al., 2019 when the day-to-day variability was analyzed combining spatial and temporal filters into one multi-dimensional retrieval. This is also an aspect for planetary waves and lidar observation as demonstrated by Eixmann et al., 2020 (AG), which is relevant for summer–winter comparison at the Arctic/Antarctic.*

As mentioned above, the considerations on polar day - polar night differences are skipped in the revised introduction, whereas comments on the uncertainties in the GWPED due to the effect of tides are included; again, the GWPED measurements of Baumgarten et al. (2017) might be uncertain but interesting enough to motivate the present paper; the potential relevance of ozone-gravity wave coupling on daylight-nighttime differences and polar day – polar night differences, and other aspects due to critical level filtering, source variability, and varying conditions are now discussed in section 3 (introduction: ll. 51-62, section 2.2.4: ll. 423-438, section 3: ll. 493-530 and ll. 551-554).

*Critical level filtering:*

*The reviewer strongly disagrees with the statement in the replies that "Critical level filtering occurs during strong westerlies between April and October (see Figure 7 of Kaifler et al., 2015)". Critical level filtering is present at all times and during all seasons, however, depending on the sign of the stratospheric winds different gravity waves encounter the critical level depending on their propagation direction and phase speed. This is directly related to the source questions and multi-step-vertical coupling processes.*

As mentioned in the second reply, the first reply was clearly related to the change from westerlies to easterlies, which is undoubtedly the most important factor in the seasonal cycle of critical level filtering. Of course, critical level filtering is present during summer months; however, this does not change the essential results of the paper which examines the principal effect of ozone-gravity wave coupling. A comment on critical level filtering is included in the discussion (ll. 515-516).

*Tidal amplitudes and gravity wave amplitudes:*

*Tidal amplitudes (semidiurnal or diurnal) can reach up to 8-15K (stratosphere/lower mesosphere) and occasionally 20 K (mesosphere) at the middle atmosphere between 30-80 km (e.g., from MERRA2). However, the amplitudes of tides are altitude-dependent and undergo the same exponential growth as gravity waves. The reviewer does not agree and has not seen observational evidence for an order of magnitude difference between tidal and gravity wave amplitudes (in a statistical sense) at the stratosphere and mesosphere. None of the lidar observations that are presented in the motivation are even close to the Rocky Mountains.*

Note again that, in the first comment to the preprint, reviewer #1 gave a general statement that tides *have almost similar or larger amplitudes compared to gravity waves at the stratosphere and mesosphere*, together with the hint on Baumgarten and Stober (2019) who found amplitudes of tides in the upper stratosphere/lower mesosphere in the order of 1 K to 2K, which is not almost larger than GW amplitudes. The example of the Rocky Mountains was only used as one counterexample to this general statement.

However, as mentioned above, planetary-scale atmospheric tides can indeed modulate the background conditions for mesoscale GWs; in the revised manuscript, the related modulation of ozone-gravity wave coupling is considered, based on the tidal amplitudes shown in the recommended paper of Baumgarten and Stober (2019) but also assuming much larger modulations, together with a note in section 3 (ll. 430-438, ll. 551-554).

*Sinusoidal approximation of gravity wave:*

*The reply draws an analogy between a gravity wave and a pendulum. This approximation seems to be by far too idealized as it skips key properties of a wave for real atmosphere application as they are found in observations. Gravity waves have a 3-dimensional wave vector and an intrinsic period and often occur not as an isolated plane wave but in wave packages. These packages have an envelope function, which is often assumed/approximated to be Gaussian. Depending on the background flow and the properties of the wave trains in the package cancelation effects are likely as updraft and downdraft phases can mix for a fixed observer on the ground in the Eulerian frame of reference. A pure vertical 1 D approximation is fine as a theoretical approach, but hard to be generalized in a real environment.*

As mentioned in the second reply, the simple image of a pendulum in the first reply was only used to clarify that the discussed effect of ozone-temperature coupling amplifies both the maximum and minimum of the oscillating wave pattern, but it does not play any role in the

paper. In the paper, the specified gravity waves are described as usual by an amplitude, a 3-dimensional vector, and an intrinsic period (now at l. 129, l. 214).

The paper uses not only one but a variety of single waves within the range of mesoscale GWs. This idealized approach is useful to achieve evidence whether the effect of ozone-gravity wave coupling is a relevant factor in the middle atmospheric circulation. For clarification, the text of the manuscript is improved (ll. 39-41, ll. 493-504, ll. 511-512, ll. 526-530).

Further investigations are needed to examine this effect in case of upward propagating gravity wave packages, particularly based on GW resolving numerical models with interactive ozone photochemistry, which is beyond on the scope of the paper; in the revised manuscript, this perspective is somewhat more highlighted (ll. 501-503, ll. 528-530).

*In summary:*

*The reviewer values the theoretical approach presented in the manuscript but has serious concerns about the motivation and justification of its importance. A revision of this manuscript either requires dealing with all the observations in more detail, including atmospheric tides and other dynamical effects as well as their biases, or skipping the observations to a large extent and just presenting the results as an idealized theoretical approach that requires observational justification. The way how the amplitude growth is well-founded between theory and observations is not appropriate. However, a justification could be also achieved by performing ICON model runs with high resolution to investigate the presented approach with resolved gravity waves in more detail. In principle, this is also possible with HIAMCM. Gravity wave resolving models permit a less ambiguous wave characterization. Such model runs will certainly strengthen the presented conclusions if confirmed. However, the reviewer understands that the model runs are a lot of work and might be postponed to future work.*

The first part of the introduction summarizing the motivation is revised, where the considerations on observations done in the preprint are skipped to a large extent; instead, the potential relevance of the analyzed effect is discussed in section 3, just presenting the results in the context of observations, and not without emphasizing that this process is one of others, and that more investigations are needed to fully understand its effects in the real atmosphere or in GW resolving numerical model simulations (ll. 51-62, ll. 493-530).

As mentioned in the second reply, I am going to perform UA-ICON model simulations with high resolution and interactively coupled ozone chemistry (with the help of some collaborators); however, this will need some time in terms of carrying out the simulations and analyzing the details of GW activity and GWPED, which might be not easier than analyzing observed temperature fluctuations; an additional note on this perspective is included in section 3 (ll. 501-503, ll. 528-530).

Reply to the comments RC2 of Referee #2 on the manuscript acp-2021-1066

Thanks again to Referee #2 for critical comments. In addition to the first reply in the open discussion, please find here the final reply including the specified changes in the revised manuscript (again, for orientation, the comments of Referee #2 are included in *Italic*).

General reply (summary)

The layout of the manuscript (section 2) is revised separating more clearly method and results.

The presentation of the concept is improved (both in the introduction and in section 2) – it might be now evident that both the ascent and descent of a sinusoidal gravity wave are amplified (i.e., the amplitude is increasing while the frequency is decreasing).

A sensitivity test on using a height-dependent or a constant scale height is included; this leads to a change in the effect of ozone-gravity wave coupling in the order of 10% to 20%.

A note on a possible relative increase of the GWPED between stratosphere and mesosphere in relation to the different seasonal cycle in stratospheric and mesospheric GW activity derived from local measurements is included in the discussion.

*The paper presents a possible mechanism of amplitude amplification of gravity waves by the interaction between ozone and gravity waves in the upper stratosphere/lower mesosphere. The paper is divided into three parts: an introduction, a section on the interaction between ozone and gravity waves, and a section titled "Summary and Conclusions." There are 6 Figures that present the results. I had difficulty following the content of the paper for several reasons.*

*First, the paper is written very compactly. The derivation of the main equations proving the positive feedback of the ozone-gravity wave coupling uses components from different sources, and I would have liked a clearer separation to make it easier for the reader. Also a clear distinction between methodology and results would be most welcome! Therefore, I propose to revise the layout of the manuscript and make it clearer.*

In the revised manuscript, sections 2.1 and 2.2 include now subsections (2.1.1 to 2.1.4, 2.2.1 to 2.2.4), separating somewhat clearer the different steps of the methodology and the results.

Further, a paragraph describing a part of the methodology (daylight-nighttime conditions for mid- and equatorial latitudes related to Figure 4d) is shifted somewhat backward to include in subsection 2.2.1 (previous preprint: ll. 344-361, revised manuscript: ll. 326-343).

*The second aspect may be a misunderstanding on my part: I cannot accept the dynamical concept of assumed gravity wave-ozone coupling (heating rate). My understanding is that propagating internal gravity waves cause positive and negative vertical displacements of the background airflow. Therefore, air transported through a gravity wave experiences both adiabatic cooling and heating. It seems to me (I found no other reference in the text) that only positive vertical velocities (i.e., displacements) are considered here to establish the "successive" or "cumulative amplitude amplification". Averaged over a horizontal wavelength*

*or one period, the net effect of gravity wave-induced cooling and warming should be zero. In conclusion, I don't see any point in publishing the results as they have been written up now. A better presentation of the underlying concept is urgently needed. Again, I could be wrong: reading the text, I would assume that gravity wave-ozone coupling leads to an increase in background temperature when gravity waves are present and ozone photochemistry is working. Is this correct? I hope, I'm right in this aspect. If not, any clarification of the dynamical concept in the paper is highly appreciated.*

Yes, of course, a sinusoidal gravity wave perturbation includes both ascent (w'>0, adiabatic heating) and descent (w'<0, adiabatic warming), and both the ascent (wave crest) and the descent (wave trough) are amplified during its propagation through the USLM; accordingly, the amplitude is increasing while the frequency is decreasing when propagating through the upper stratosphere/lower mesosphere.

For clarification, the text of the manuscript is improved from the beginning (abstract and introduction: ll., 32-34, ll. 73-83; section 2.1.2: ll. 142-146, ll. 163-165, ll. 185-187; section 2.1.4, ll. 253-254, ll. 262-264; section 3, ll. 462-464).

As mentioned in the first reply, the theoretical approach describes upward propagating gravity waves in a constant background, i.e., they cannot change anything in the background. The identified process could only lead to a change in a varying background if the stronger increase in the GW amplitudes with height would lead to a change in gravity wave breaking processes driving the middle atmospheric circulation; this potential effect is discussed as an outlook of further examinations, which are beyond the scope of the paper because extensive model calculations are needed. In the revised manuscript, this is somewhat more highlighted (abstract: ll. 39-41, and section 3: ll. 497-504 and ll. 526-530, in addition to ll. 538-540).

*There is a third point that should be considered in a new version of the manuscript. The whole gravity wave concept relies on linear wave theory. However, the authors use a density scale height H that is strictly only applicable for an isothermal atmosphere as it is constant with altitude. Already in the textbooks by Gill (1982, page 50 top) and by Dutton (1976, pages 67-68) altitude-dependent scale heights are mentioned or proposed. Recently, Reichert et al. (2021) used a height-dependent H for investigating conservative growth rates from ground-based lidar measurements. So, it would be worthwhile to estimate the amplitude growth in an atmosphere with temperature varying with altitude. Especially, in the summer mesosphere where the temperatures can drop drastically from the stratopause to the cold mesopause, this effect might account for some of the observed exponential increase.*

Yes, the choice of the scale height H can affect the change in the amplitude growth due to ozone-gravity wave coupling especially in the summer mesosphere. For clarification, the revised manuscript includes a sensitivity test as summarized as follows.

In the present paper, the solutions of the amplitude growth are formulated on pressure levels, therefore the height-dependence of $H=H(T_0)$ is included in the cumulative level-by-level amplification; for clarification, this is explicitly mentioned in the revised manuscript, including a hint on the sensitivity test using a constant scale height $H_0$ (section 2.1.4: ll. 249-250, section 2.2.1: ll. 307-312, and ll. 319-321).

The sensitivity test is realized by setting $H_0$=7 km in the specification of the distance between the levels used in the calculation of the cumulative amplitude amplification. As expected, this leads to a significant change in the upper mesospheric GW amplitudes (up to 10%) and GWPED (up to 20%) particularly at summer polar latitudes, where $H=H(T_0)$ varies in the USLM region between 6.5 km and 7.5 km (i.e., less than 10%); however, this modulation is less than the first-order process of ozone-gravity wave coupling (details of the sensitivity test and a conclusive note are included in subsection 2.2.4, ll. 439-452, and section 3, ll. 555-559).

*Last but not least, I see an essential difference in the gravity wave regimes of the upper stratosphere and lower mesosphere between summer and winter. This picture results from Figure 6 of Reichert et al (2021): it shows almost no seasonal variability of Ep in the layer 65 to 80 km altitude in contrast to the layers below. Thus, the mesosphere seems to be a region where gravity waves always exist almost independent from the local excitation at the place of the observations. Where these waves come from, if they are from primary or secondary or other sources, I don't know but they seem to be present all the time. In conclusion, the strong summer increase can probably also be explained by the reduced local excitation conditions, i.e. the strongly reduced Ep values at lower layers. Sure, this is for one location in the lee of the Andes but it is a convincing example. By the way, there is a further aspect not discussed in the paper: the superposition of gravity waves from different sources entering the observational volume horizontally and leading to enhanced Ep values as indicated by Reichert et al. (2021) as well.*

Thanks again for the comment, and the hint on the publication of Reichert et al. (2021). A related comment on the stratospheric GW sources and the relative relation between mesospheric and stratospheric GWPED is included (section 3, ll. 510-511, ll. 520-526).

Please note again that the paper does not want to explain all the details of published GWPED profiles. The aim of the paper is to demonstrate that the process of ozone-gravity wave coupling might principally lead to significant GW amplitude amplifications and daylight-nighttime differences in the GWPED, as earlier suggested by Baumgarten et al. (2017). The results of the theoretical approach are in the order of observed relative relations between mesospheric and stratospheric GWPED values, therefore it is justified to conclude that this process is a relevant factor in the summer middle atmosphere, although the addressed process but also a lot of other processes play an important role for understanding the total GWPED at a specific location. The manuscript is revised from the beginning to make this point as clear as possible (abstract: ll. 28-31, ll. 39-40, introduction: ll. 51-62, section 3: ll. 493-530).

*I would have liked to see the authors pay more attention to these possible dynamical aspects and their potential impact on growth rates. A discussion of both the dynamical and ozone temperature aspects would improve the paper and relate its new results to known published knowledge.*

The addressed aspects are included in the revised manuscript. The discussion is extended including some additional references. Also, some sensitivity calculations are added to provide information on the involved dynamical and ozone-temperature coupling aspects (see particularly subsection 2.2.4: ll. 423-452, section 3: ll. 493-530 and ll. 551-559).

*Minor Comments:*

*line 48: "over-exponential" is probably not well-selected as term: what does it mean? I guess, you refer to exponential growth with a enhanced rate, correct?*

The terminus "over-exponential" is deleted in the whole manuscript.

*line 79-80: here, the concept of w'>0 is introduced for the first time. I thought, well, why do the author not consider w'<0 as well as vertical displacements related to these vertical oscillations vary in time and space regularly in a gravity wave.*

Of course, the description is valid for both w'>0 and w'<0. The manuscript is improved from the beginning to make this clearer (e.g., abstract and introduction: ll., 32-34, ll. 73-83; see also the reply to your second aspect above).

*line 114: introduce minus sign in density equation*

Done. Thank you.

*line 115: why is v_0 d/dy missing in the total derivative?*

Done. Thank you.

*line 238: Figure 8 of Reichert et al. (2021) shows that the majority of vertical wavelengths is about and large than 15 km. So, the choice of the selected parameters (especially with reference to the Andes) is not clear to me.*

The selected parameters are used as examples of GW characteristics where ozone-gravity wave interaction is particularly efficient (as shown in Figure 3). The reference to the Andes is deleted because it is not necessary in this subsection (the sentence is now at ll. 241).

Considering the orography of the Andes (horizontal extension and height), I would expect orographically forced GWs with horizontal wavelengths of several hundred km and vertical wavelengths of around 3 km to 5 km, travelling over the South Atlantic in case of the usual westerlies; therefore, it was especially mentioned. In the revised manuscript, the potential physical GW sources including the Andes are mentioned in the discussion (l. 469, l. 562).

*line 266: Why do you use "but" not "and"?*

Done. Thank you.

*References:*

*Dutton, J. A., 1976: The Ceaseless Wind. 1st ed., McGraw-Hill, New York and London, 579 pp.*

*Gill, A. E., 1982: Atmosphere-Ocean Dynamics, Academic Press, 1st edn.,662 pp.*

*Reichert, R. et al. 2021: High-cadence lidar observations of middle atmospheric temperature and gravity waves at the Southern Andes hot spot. Journal of Geophysical Research: Atmospheres, 126, e2021JD034683. https://doi.org/10.1029/2021JD034683*

---

## Referee Report (RR1)

Second review on:

Ozone-Gravity Wave Interaction in the Upper Stratosphere/Lower Mesosphere

by

Axel Gabriel

General Comment:
The reviewer appreciates the efforts undertaken to prepare a revised manuscript. Unfortunately, the revised manuscript did not pick up some of the suggestions and even contains statements that are questionable. The reviewer provides below a detailed list of reasons why the manuscript in its present form is suggested for rejection. In parts, some of these aspects originate from different viewpoints between theoretical approaches and experimental data analysis and the current understanding of vertical coupling processes and so forth. Some simplifications are justifiable for theoretical solutions as the proposed ozone-gravity interaction, but a simple relation to observations is not adequate to support the conclusions. This is really a problematic aspect of the revised manuscript.

Main concerns:
Lidar data interpretation:
The manuscript is now motivated entirely by one single lidar paper (Baumgarten et al., 2017). The main narrative of the paper is a gravity wave lidar climatology leveraging a daylight capable Rayleigh lidar at the mid-latitudes.  Although there is one statement in the conclusion about the day and night differences associated with atmospheric tides and the corresponding filtering, it is only speculated on these differences. During daylight, these lidars show an increased noise floor due to the sunlight, which causes a less good signal-to-noise ratio compared to nighttime measurements, which results in larger or increased fluctuations for the analyzed temperatures. However, this increased variability is mainly the result of hydrostatic integration applied for the temperature inversion and not a sign of geophysical variability. Rüfenacht et al., 2018 show an impressive example of the day and night noise levels in the sister Rayleigh system at Andenes.

Furthermore, even in Baumgarten et al., 2017 there is no convincing signature of such a day and night difference (see Figure 3).

Atmospheric tides:

Although it appears eligible to use HARMONIA without tides as background for the theoretical prediction of the ozone-gravity wave interaction, it is not advisable to ignore tides in the lidar data. Most lidar soundings are significantly biased due to the short record lengths and proper removal of tidal effects is often not possible in the GWPED. Most published amplitudes of individual tidal modes are in the order of a few K, which is often related to a systematic underestimation from the applied superposed epoch analysis. Just looking at Figure 1 in Baumgarten et al., 2019 or attached MERRA2 data reveals a temperature difference between 50-60 km of about 30 K, which corresponds to a lapse rate of -3 K/km. This value is 20-30 times larger than the 0.1 K/km stated in the manuscript, which means that all the discussion and included factors are obsolete. In fact, all factors are increased by a factor 20-30 and, thus, become no longer negligible. The reviewer assumes that the small vertical temperature gradient was estimated from the vertical profile of the mean tidal amplitudes, which is misleading in this case.

Below there are two panels showing data from MERRA2 for an undisclosed mid-latitude location. The dominating feature is the diurnal tide in the wind and temperature.

[Figure]

[Figure]

undisclosed mid-latitude location MERRA2 2018 UTC

The corresponding gravity wave activity for the same period is shown below. The mean and atmospheric tides have been removed. The waves discussed by the manuscript are partially resolved from MERRA2 and appear as the coherent structure at altitudes from 16-50 km. Above the coherence disappears, and a superposition of upward and downward phase lines becomes evident. However, it is not clear whether the data assimilation 3DVAR in MERRA2 captures the secondary wave generation around 60 km (fishbone structure Vadas et a., 2018a,b) or whether these waves are reflected from the model top and not sufficiently suppressed by the sponge layer.

[Figure]

undisclosed mid-latitude location MERRA2 2018 UTC

The reviewer is not convinced that the arguments listed in lines 425-438 actually are valid looking at the MERRA2 data, which is also confirmed by other meteorological reanalysis and model fields. Furthermore, the gravity residuals reflect almost no coherence above 55-60 km, which also does not support the proposed effect considering that a sun-synchronous phase relation of the gravity waves would be required to sustain a fixed phase relation to the ozone to ensure the day and night differences.

Background winds and atmospheric waves:
Reanalysis data, as well as observations, indicate that the gravity wave amplitudes are strongly affected depending on the phase velocity and direction of the gravity wave relative to the background winds. At the altitudes presented and discussed in the submitted manuscript, these changes are mostly driven by atmospheric tides with rather short vertical wavelengths and, thus, a sudden amplitude growth of the gravity waves is also explainable just by changes in the background winds, which are not captured or considered in the theoretical framework.

Multistep vertical coupling:
As already mentioned, the HIAMCM model, as well as reanalysis data including MERRA2, reveal fishbone structures, which are associated with local body forces of breaking gravity waves or jet-induced instabilities (Vadas et al., 2018a,b, and many others). The GWPED observations are not discussed whether the increase in amplitude could be caused by that effect. The manuscript makes an attempt to justify the amplification only by the ozone effect, which seems unlikely and needs to be quantified. In summary, the cited GWPED data is not supporting the conclusions of the theoretically predicted amplification.

Recommendation:
The reviewer suggests minimizing the lidar part to the introduction and as motivation, but clearly describing that the lidar data does not permit to distinguish between the nature or source of the gravity waves and whether multistep vertical coupling or increased noise during the daylight could explain the tiny anomalies in GWPED as well.
An experimental and convincing case would be to search for a resolved large-scale sun-synchronous wave in MERRA2 or HIAMCM and to run the ray-tracer GROGRAT to track the wave and its amplitude to search for a second case during nighttime to demonstrate the opposite behavior. However, it might

already be helpful to identify gravity waves in ozone data to show that at least the ozone shows some response.

References:

Vadas, S. L., Zhao, J., Chu, X., & Becker, E. (2018). The excitation of secondary gravity waves from local body forces: Theory and observation. *Journal of Geophysical Research: Atmospheres*, *123*, 9296–9325.https://doi.org/10.1029/2017JD027970

Vadas, S. L., & Becker, E. (2018). Numerical modeling of the excitation, propagation, and dissipation of primary and secondary gravity waves during wintertime at McMurdo Station in the Antarctic. *Journal of Geophysical Research: Atmospheres*, *123*, 9326–9369. https://doi.org/10.1029/2017JD027974

Rüfenacht, R., Baumgarten, G., Hildebrand, J., Schranz, F., Matthias, V., Stober, G., Lübken, F.-J., and Kämpfer, N.: Intercomparison of middle-atmospheric wind in observations and models, Atmos. Meas. Tech., 11, 1971–1987, https://doi.org/10.5194/amt-11-1971-2018, 2018.

---

## Author Response (AR2)

Reply to the second reviews of Referees #1 and #2 on the manuscript acp-2021-1066 (Ozone-gravity wave interaction in the upper stratosphere/lower mesosphere, by A. Gabriel)

1) Reply to the comments of Referee #1

Many thanks again to Referee #1 for critical comments. Please find here the reply (again, for orientation, the comments of Referee #1 are included in *Italic*).

*General Comment:*

*The reviewer appreciates the efforts undertaken to prepare a revised manuscript. Unfortunately, the revised manuscript did not pick up some of the suggestions and even contains statements that are questionable. The reviewer provides below a detailed list of reasons why the manuscript in its present form is suggested for rejection.*

I am a bit irritated, because I do not really see that I did not pick up some of the earlier suggestions. However, the manuscript is revised again carefully following the list of main concerns given below.

*In parts, some of these aspects originate from different viewpoints between theoretical approaches and experimental data analysis and the current understanding of vertical coupling processes and so forth. Some simplifications are justifiable for theoretical solutions as the proposed ozone-gravity interaction, but a simple relation to observations is not adequate to support the conclusions. This is really a problematic aspect of the revised manuscript.*

I do not really understand these statements. The theoretical approach of the paper is very clear: it is based on standard equations describing gravity waves in a constant background flow, with an additional equation for the ozone perturbations due to gravity waves but excluding other processes to provide clear understanding of the potential effect on the GW amplitudes (such an approach is usual before introducing a sophisticated parameterization into a model); the resulting differences between GW amplitudes with and without ozone-gravity waves gives an idea of this potential effect in comparison with cited GWPED values already published in refereed journals by other authors; the paper does not include any experimental data analysis, and it does not want to explain entirely the GWPED values derived from measurements *only* by ozone-gravity wave interaction or to give a simple relation to observations. In the revised manuscript, this is somewhat more highlighted from the beginning (abstract: ll. 28-30, introduction: ll. 86-89, discussion: ll. 553-555, ll. 571-574).

Obviously, the paper can still lead to some misunderstanding, perhaps because only single GWs are considered, with focus on the relative change in the GW amplitudes. Therefore, an additional Figure 7 is added illustrating the amplification of the GWPED for mean values averaged over representative mesoscale GWs with different horizontal and vertical wavelengths, which gives an additional idea on the potential effect (abstract: ll. 39-42; new section 2.2.5, ll. 483-509; discussion: ll. 544-555). Figure 7 includes not only the relative but also the absolute amplitude amplifications for moderate initial GW perturbations in the middle stratosphere (1 K), which might illustrate somewhat clearer that this effect is not *tiny* as claimed by reviewer #1 below.

*Main concerns:*

*Lidar data interpretation:*

*The manuscript is now motivated entirely by one single lidar paper (Baumgarten et al., 2017). The main narrative of the paper is a gravity wave lidar climatology leveraging a daylight capable Rayleigh lidar at the mid-latitudes. Although there is one statement in the conclusion about the day and night differences associated with atmospheric tides and the corresponding filtering, it is only speculated on these differences. During daylight, these lidars show an increased noise floor due to the sunlight, which causes a less good signal-to-noise ratio compared to nighttime measurements, which results in larger or increased fluctuations for the analyzed temperatures. However, this increased variability is mainly the result of hydrostatic integration applied for the temperature inversion and not a sign of geophysical variability. Rüfenacht et al., 2018 show an impressive example of the day and night noise levels in the sister Rayleigh system at Andenes.*

The manuscript is improved again to highlight that the cited daylight-nighttime differences are only one point of Baumgarten et al. (2017), and that these differences include uncertainties due to a less good signal-to-noise ratio during daylight (abstract: ll. 28; introduction, ll. 75-78). Please note again that the conclusion of Baumgarten et al. (2017) concerning the daylight-nighttime differences is cited accurately as published in a refereed journal, and that these findings were only one motivation of others. The aim of the present paper is to examine the potential effect of ozone-gravity wave interaction excluding other processes, and not to explain entirely total GWPED values derived from some few specific Lidar measurements. This is somewhat more highlighted from the beginning (see reply to general comment above).

*Furthermore, even in Baumgarten et al., 2017 there is no convincing signature of such a day and night difference (see Figure 3).*

This statement is not right. A comparison between GWPED values derived from full-day and nighttime observations are given in Figure 6 and Figure 9 of Baumgarten et al. (2017). For July, the GWPED values are significantly stronger during full-day- than nighttime observations by a factor of approximately 2 (significant within one standard deviation). For clarification, a related comment is included in the revised manuscript (ll. 70-74).

*Atmospheric tides:*

*Although it appears eligible to use HARMONIA without tides as background for the theoretical prediction of the ozone-gravity wave interaction, it is not advisable to ignore tides in the lidar data. Most lidar soundings are significantly biased due to the short record lengths and proper removal of tidal effects is often not possible in the GWPED. Most published amplitudes of individual tidal modes are in the order of a few K, which is often related to a systematic underestimation from the applied superposed epoch analysis.*

Yes, I agree, it is not advisable to ignore tides when interpreting fluctuations derived from lidar data. Interpreting the fluctuations derived from lidar data has been done by those authors who presented the cited GWPED values as a reliable result of lidar measurements in refereed journals, including uncertainty ranges because of the difficulties in separating the GW

perturbations and the fluctuations due to tides. In the present paper on ozone-gravity wave interaction, these GWPED values derived from lidar data are cited because they give a benchmark and orientation on the GWPED values in the stratosphere and mesosphere, nothing else (see revised introduction, ll. 53-67). Please note here again that the purpose of the present paper is to estimate the potential effect of ozone-gravity wave interaction on the daylight-nighttime differences in GW amplitudes as clear as possible excluding other processes like tides (ll. 86-89, ll. 315-316).

Uncertainties of the GWPED values derived from lidar data are already highlighted in the introduction, particularly the problem of temporal filtering methods and speculations whether diurnal or semidiurnal tides could contribute to the daylight-nighttime differences in the GWPED values derived by Baumgarten et al. (2017) (ll. 74-80). Possible effects of the modulation of the background conditions due to tides are already discussed (ll. 452-467). In the revised lidar part in the discussion, the possible role of tides for the daylight-nighttime differences is somewhat more highlighted (ll. 607-618).

*Just looking at Figure 1 in Baumgarten et al., 2019 or attached MERRA2 data reveals a temperature difference between 50-60 km of about 30 K, which corresponds to a lapse rate of -3 K/km. This value is 20-30 times larger than the 0.1 K/km stated in the manuscript, which means that all the discussion and included factors are obsolete. In fact, all factors are increased by a factor 20-30 and, thus, become no longer negligible. The reviewer assumes that the small vertical temperature gradient was estimated from the vertical profile of the mean tidal amplitudes, which is misleading in this case.*

I do not understand these estimations. Figure 1 of Baumgarten and Stober (2019) show total temperatures at mid-latitudes during May, illustrating the total lapse rate. In the manuscript, the total temperature and the related time-mean lapse rate are prescribed as background with a difference between 50-60 km of about 30 K at summer mid-latitudes produced by the HAMMONIA model (see Figure 1a, ll. 121-122, ll. 258-261). Figure 2 of Baumgarten and Stober (2019) show tidal variations between $-5$ K and $+5$ K and the following figures of Baumgarten and Stober (2019) show the amplitudes of the tidal variations with increase from about 0.5 K at 30 km up to about 4 K at 70km, derived from lidar data and MERRA2. This variation of the background lapse rate in the order of 0.1 K/km is used as input in the sensitivity test described in section 2.2.4 (ll. 459-465). Even an artificially assumed much larger change in the lapse rate (10% to 50% instead of 1%, corresponding to $-1$ to $-5$ K/km) does not change the amplification of the upper mesospheric GW amplitudes by more than 10% (ll. 465-467). This relatively weak sensitivity of the amplification to variations in the lapse rate $\gamma$ is related to the facts that the stability in terms of $N_0^2 = g\,(\Gamma - \gamma)/T_0$ is primarily given by the dry adiabatic lapse rate $\Gamma \approx 10$ K/km and that the introduced ozone adiabatic lapse rate in terms of $N_\mu^2$ is a linear function of $N_0^2$ (see Eq. (14)). Therefore, the discussion and the derived factors are not obsolete.

*Below there are two panels showing data from MERRA2 for an undisclosed mid-latitude location. The dominating feature is the diurnal tide in the wind and temperature.*

*The corresponding gravity wave activity for the same period is shown below. The mean and atmospheric tides have been removed. The waves discussed by the manuscript are partially resolved from MERRA2 and appear as the coherent structure at altitudes from 16-50 km. Above the coherence disappears, and a superposition of upward and downward phase lines becomes evident. However, it is not clear whether the data assimilation 3DVAR in MERRA2 captures the secondary wave generation around 60 km (fishbone structure Vadas et a.,*

*2018a,b) or whether these waves are reflected from the model top and not sufficiently suppressed by the sponge layer.*

*The reviewer is not convinced that the arguments listed in lines 425-438 actually are valid looking at the MERRA2 data, which is also confirmed by other meteorological reanalysis and model fields. Furthermore, the gravity residuals reflect almost no coherence above 55-60 km, which also does not support the proposed effect considering that a sun-synchronous phase relation of the gravity waves would be required to sustain a fixed phase relation to the ozone to ensure the day and night differences.*

Thanks a lot for the figures. As mentioned in the previous reply, the secondary GW structures are indeed interesting, but I do not really understand why this should be discussed in detail in the present paper. The assimilation model of MERRA2 does not include ozone-gravity wave interaction in the upper stratosphere/lower mesosphere because the photochemistry module uses monthly 2-dimensional ozone production rates and loss frequencies derived from a two-dimensional chemistry model (Stajner et al., 2008; Wargan et al., 2015; this approach is sufficient for the purposes of the MERRA2 assimilations because ozone has a sufficiently long lifetime in the lower and middle stratosphere). Therefore, MERRA2 data seems to be not suitable to support or to reject the presented results on ozone-gravity wave interaction.

Stajner I, et al. Assimilated ozone from EOS-Aura: Evaluation of the tropopause region and tropospheric columns. J Geophys Res. 2008;113:D16S32. doi: 10.1029/2007JD008863.

Wargan K, Pawson S, Olsen MA, Witte JC, Douglass AR, Ziemke JR, Strahan SE, Nielsen JE. The global structure of upper troposphere- lower stratosphere ozone in GEOS-5: A multiyear assimilation of EOS Aura data. J Geophys Res Atmos. 2015;120:2013–2036. doi: 10.1002/2014JD022493.

Perhaps I may add that I see in the plots not only fishbone structures but also some coherent GW structures with increasing amplitude above 50 km, some up to 70 km. It is known that dissipation increases if the GW amplitudes increase with height, which might be the reason that a fraction of the coherent GW structures disappear. The origin of the GW structures above 60 km might be unclear not only because of reflection from the model top but also because of artificial non-geostrophic flow components forced by setting the upper boundary of the atmosphere at the model top. However, I agree with reviewer #1 that the GW structures above 60 km produced by the MERRA2 assimilation model are very uncertain; therefore, here again, they are not suitable to support or to reject the results of the present paper.

However, the concern of referee #1 might be related to short-term fluctuations due to tides and not to a modulation of the large-scale background estimated in section 2.2.4 (lines 425-438 are now at ll. 454-467). As mentioned above (and in the paper), short-term fluctuations in the balanced winds due to tides or other processes are explicitly excluded by the approach to focus on the effect of ozone-gravity wave interaction as clear as possible. However, in the revised manuscript, possible effects of these fluctuations and related nonlinear interactions with GWs are now more highlighted in the discussion (ll. 595-618; see also the reply to the related main concerns on *background winds* and *multistep vertical coupling* below).

*Background winds and atmospheric waves:*

*Reanalysis data, as well as observations, indicate that the gravity wave amplitudes are strongly affected depending on the phase velocity and direction of the gravity wave relative to*

*the background winds. At the altitudes presented and discussed in the submitted manuscript, these changes are mostly driven by atmospheric tides with rather short vertical wavelengths and, thus, a sudden amplitude growth of the gravity waves is also explainable just by changes in the background winds, which are not captured or considered in the theoretical framework.*

The present paper assumes the usual standard approach of upward propagating GWs in a slowly varying background flow excluding short-term fluctuations of the background wind due to atmospheric tides or other processes (ll. 86-89, ll. 315-316). In other words, yes, of course, changes in the amplitude growth of gravity waves are not captured or considered by the theoretical framework because they are explicitly excluded from the beginning to estimate the effect of ozone-gravity wave interaction as clear as possible. Usually, such an approach assuming GWs in a slowly varying background is the starting point of GW parameterizations used in current general circulation models; therefore, the results of the paper could be quite interesting for many researchers.

However, slowly varying changes in the background wind are discussed in section 2.2.4 (ll. 459-465), whereas a quantification of the effects of short-term changes in the balanced winds (short-term changes in the background winds do not exist because they are not a slowly varying background for the short-term perturbations), and associated non-linear interaction between the upward propagating GWs and the short-term fluctuations of the balanced winds, needs much more sophisticated solutions or extensive numerical simulations with sufficiently high spatial and temporal resolution (to my knowledge this is an issue of current research), in case of examining ozone-gravity interaction with including an interactively coupled photochemistry model, which is beyond the scope of the paper. This is somewhat more highlighted in the discussion (ll. 595-618).

*Multistep vertical coupling:*

*As already mentioned, the HIAMCM model, as well as reanalysis data including MERRA2, reveal fishbone structures, which are associated with local body forces of breaking gravity waves or jet-induced instabilities (Vadas et al., 2018a,b, and many others). The GWPED observations are not discussed whether the increase in amplitude could be caused by that effect. The manuscript makes an attempt to justify the amplification only by the ozone effect, which seems unlikely and needs to be quantified. In summary, the cited GWPED data is not supporting the conclusions of the theoretically predicted amplification.*

Both the HIAMCM model and the MERRA2 assimilation model do not include the feedback of ozone photochemistry to GW perturbations, therefore these models are not suitable to provide any conclusion on ozone-gravity wave interaction. The fishbone structures and the related processes are indeed interesting, but it is not the purpose of the manuscript to examine and to quantify the possible effects of these processes on observed GWPED observations. I guess these fishbone structures will change significantly if introducing ozone-gravity wave coupling into the models; however, this would be an issue of another paper.

As already mentioned in the first reply, it is not right to claim that the manuscript wants to explain the daylight-nighttime differences in the GWPED *only* by the ozone effect. The purpose of the manuscript is to examine and to quantify the effect of ozone-gravity wave interaction excluding other processes, and then to have a look whether the resulting amplification is strong or weak in relation to total GWPED values published in refereed

journals. If it is strong enough, the conclusion that ozone-gravity wave interaction is one significant component amplifying the GW amplitudes and GWPED values is justified. This usual procedure is meaningful before introducing this process in sophisticated GWD parameterizations used in circulation models or in extensive numerical model calculations with high spatial and temporal resolution and interactive photochemistry.

The manuscript is revised again to make this point clearer (ll. 53-89; ll. 539-555; see also reply to general comment above). An additional comment which explicitly highlights that also other processes like multistep vertical coupling could principally contribute to the daylight-nighttime differences is included in the discussion (ll. 595-618).

*Recommendation:*

*The reviewer suggests minimizing the lidar part to the introduction and as motivation, but clearly describing that the lidar data does not permit to distinguish between the nature or source of the gravity waves and whether multistep vertical coupling or increased noise during the daylight could explain the tiny anomalies in GWPED as well.*

Introduction and discussion are improved (ll. 53-89, ll. 539-555, ll. 566-574, ll. 595-618). I never claimed that any *lidar data permit to distinguish between the nature or source of the gravity waves* – I am still a bit irritated and surprised about such a statement.

A comment on increased noise during daylight is included in the introduction (ll. 75-79). A comment on the possible role of multistep vertical coupling is included in the discussion (ll. 595-618). The potential effect of ozone-gravity wave coupling on the absolute daylight-nighttime differences in the GWPED is not tiny in comparison to observed values, as now somewhat more clearly illustrated and discussed with the help of a new Figure 7 (see also reply to general comment above).

*An experimental and convincing case would be to search for a resolved large-scale sun-synchronous wave in MERRA2 or HIAMCM and to run the ray-tracer GROGRAT to track the wave and its amplitude to search for a second case during nighttime to demonstrate the opposite behavior. However, it might already be helpful to identify gravity waves in ozone data to show that at least the ozone shows some response.*

Generally, a theoretical approach like that of the present paper is quite convincing because of the evidence of the analytic solutions, and a helpful preliminary for performing new measurements or numerical simulations. Using current versions of the models suggested by reviewer #1 is not convincing because the feedback of GW perturbations to ozone production and loss rates are not included. Perhaps (or hopefully) the paper will stimulate further model developments including this process; however, this is beyond the scope of the present paper, as already mentioned in the discussion (ll. 582-584, ll. 608-615).

Yes, I agree, it would be very helpful to identify gravity waves in ozone data for the altitude range of the upper stratosphere/lower mesosphere region, together with identifying GWs in simultaneously measured temperatures. It is well known that gravity wave structures can be found in ozone but also in other trace gas constituents like methane or stratospheric aerosol in the lower and middle stratosphere; however, to my knowledge, suitable data sets with

sufficiently high vertical resolution do not exist for the USLM region. A related note for stimulating such measurements is already included at the end of the discussion (ll. 619-622).

*References:*

*Vadas, S. L., Zhao, J., Chu, X., & Becker, E. (2018). The excitation of secondary gravity waves from local body forces: Theory and observation. Journal of Geophysical Research: Atmospheres, 123, 9296–9325.https://doi.org/10.1029/2017JD027970*

*Vadas, S. L., & Becker, E. (2018). Numerical modeling of the excitation, propagation, and dissipation of primary and secondary gravity waves during wintertime at McMurdo Station in the Antarctic. Journal of Geophysical Research: Atmospheres, 123, 9326–9369. https://doi.org/10.1029/2017JD027974*

*Rüfenacht, R., Baumgarten, G., Hildebrand, J., Schranz, F., Matthias, V., Stober, G., Lübken, F.-J., and Kämpfer, N.: Intercomparison of middle-atmospheric wind in observations and models, Atmos. Meas. Tech., 11, 1971–1987, https://doi.org/10.5194/amt-11-1971-2018, 2018.*

These references are included.

2) Reply to the comments of Referee #2

Many thanks to Referee #2 for acceptance and for the final comments. Please find here the reply (again, for orientation, the comments of Referee #1 are included in *Italic*).

*Still, there are some minor points that could be clarified and modified in the text.*

*- from time to time, "temperature" is used instead of "potential temperature" (e.g. in line 203); I suggest to check the manuscript accordingly*

Yes, thank you. The manuscript is checked and improved where necessary (see some few tracked changes in Section 2.1).

*- the transition from the Lagrangian view (the total derivatives in equations (1)-(6)) to the results presented in the text and figures as "local" changes could be clearer: local changes are partial (..)/partial t: you write equations for d_0 (..)/dt. Maybe it's something quite obvious, but I always get stuck when I read these lines of results. Do you consider the total effect on Q and O3 that an air parcel experiences when traveling along a sinusoidal trajectory through the wave (integrated over one cycle or wavelength)?*

Yes, indeed, this could be misunderstanding. The terminus "local" means here the amplification at a specific level (Section 2.1) which is a preliminary to the amplification during the upward propagation (Section 2.2). This is improved throughout the whole paper (mostly "at a specific level", which is already used, instead of "local").

I consider the total change of a sinusoidal GW pattern over one cycle or wavelength travelling within the background flow, where both the ascent and the descent branch of this sinusoidal wave perturbation are amplified. This leads to an increase in the amplitude (e.g., in terms of $|T'^2|$). However, I do not integrate over a wavelength, an integration over ascent and descent of the sinusoidal wave would be zero. A net effect at a specific level would occur if the amplification of the amplitude would lead to a change in gravity wave breaking processes, which is somewhat more highlighted (ll. 508-509, ll. 552-553).